# CORESETS FOR CLUSTERING WITH NOISY DATA

## ABSTRACT

We study the problem of data reduction for clustering when the input dataset $\widehat{P}$ is a noisy version of the true dataset $P$. Motivation for this problem derives from settings where data is obtained from inherently noisy measurements or noise is added to data for privacy or robustness reasons. In the noise-free setting, coresets have been proposed as a solution to this data reduction problem – a coreset is a subset $S$ of $P$ that comes with a guarantee that the maximum difference, over all center sets, in cost of the center set for $S$ versus that of $P$ is small. We find that this well-studied measure which determines the quality of a coreset is too strong when the data is noisy because the change in the cost of the optimal center set in the case $S = \widehat{P}$ when compared to that of $P$ can be much smaller than other center sets. To bypass this, we consider a modification of this measure by 1) restricting only to approximately optimal center sets and 2) considering the *ratio* of the cost of $S$ for a given center set to the minimum cost of $S$ over all approximately optimal center sets. This new measure allows us to get refined estimates on the quality of the optimal center set of a coreset as a function of the noise level. Our results apply to a wide class of noise models satisfying certain bounded-moment conditions that include Gaussian and Laplace distributions. Our results are not algorithm-dependent and can be used to derive estimates on the quality of a coreset produced by any algorithm in the noisy setting. Empirically, we present results on the performance of coresets obtained from noisy versions of real-world datasets, verifying our theoretical findings and implying that the variance of noise is the main characterization of the coreset performances.

## 1 INTRODUCTION

Clustering problems are ubiquitous in machine learning with diverse applications (Lloyd, 1982; Tan et al., 2006; Arthur & Vassilvitskii, 2007; Coates & Ng, 2012). An important class of clustering problems is called $(k, z)$-CLUSTERINGwhere, given a dataset $P \subset \mathbb{R}^d$ of $n$ points and a $k \geq 1$, the goal is to find a set $C \subset \mathbb{R}^d$ of $k$ points that minimizes the cost $\text{cost}_z(P, C) := \sum_{x \in P} d^z(x, C)$. Here $d^z(x, C) := \min \left\{ d^z(x, c) : c \in C \right\}$ is the distance of $x$ to the center set $C$ and $d^z$ denotes the $z$-th power of the Euclidean distance. Examples of $(k, z)$-CLUSTERINGinclude $k$-MEDIAN (when $z = 1$) and $k$-MEANS (when $z = 2$). In many applications, the dataset $P$ is large, and it is desirable to have a small representative subset that requires less storage and compute while allowing us to solve the underlying clustering problem. Coresets have been proposed as a solution towards this (Har-Peled & Mazumdar, 2004) – a coreset is a subset $S$ that approximately preserves the clustering cost for all center sets. Coresets have found further applications in sublinear models, including streaming (Har-Peled & Mazumdar, 2004; Braverman et al., 2016), distributed (Balcan et al., 2013; Huang et al., 2022b), and dynamic settings (Henzinger & Kale, 2020) due to the ability to merge and compose them (see, e.g., (Wang et al., 2021, Section 3.3)).

In the majority of results on coresets for clustering, the dataset $P$ is assumed to be accurately known. Data in the real world, however, may often be *noisy*. One reason is that the measurement process may itself introduce noise in data, or corruptions may occur in the recording or reporting processes (Halevy et al., 2006; Agrawal et al., 2010; Sáez et al., 2013; Iam-on, 2020). Further, noise can be introduced intentionally in data due to privacy concerns (Ghinita et al., 2007; Dwork et al., 2014; Ghazi et al., 2020), or to ensure robustness (Ying, 2019; Li et al., 2019). In these settings, one observes a noisy dataset $\widehat{P} \subset \mathbb{R}^d$ instead of the true dataset $P$. Various types of noise can arise in different scenarios: stochastic noise, adversarial noise, and noise due to missing data (Balcan et al., 2008;

Batista & Monard, 2003; Iam-on, 2020). The knowledge of noise in data can range from completely unknown to known parameters of a known distribution. Stochastic noise, in particular, has been examined in fundamental problems such as clustering (Iam-on, 2020) and regression (Theodoridis, 2020), and is commonly observed in social sciences (Baye & Monseur, 2016; O'Dea et al., 2018; Fong et al., 2022), economics (Fang & Moro, 2010), and machine learning (Dwork et al., 2014; Ying, 2019; Li et al., 2019). When the attributes of data interact weakly with each other, independent and additive stochastic noise is considered (Zhu & Wu, 2004; Freitas, 2001; Langley et al., 1992), i.e., for every point $p \in P$ and every attribute $j \in [d]$, the observed point is $\widehat{p}_j = p_j + \xi_{p,j}$ where $\xi_{p,j}$ is drawn from a given distribution $D_j$. Various choices of $D_j$ have been considered: Gaussian distribution (Secondini, 2020; Helou & Süsstrunk, 2019), Laplace distribution (Bun et al., 2019), uniform distribution (Agrawal et al., 2010; Roizman et al., 2019), and Dirac delta distribution (Zimmermann & Dostert, 2002). Such additional noise may come from inherent individual variations, such as STEM scores (Baye & Monseur, 2016; O'Dea et al., 2018) whose mean and variance can be estimated by multiple exams; or employers making employment decisions via statistical information on the group they belong to (Fang & Moro, 2010). In privacy or robustness settings, we may also add noise with known parameters to data, e.g., the deliberate addition of i.i.d. Gaussian noise in the Gaussian mechanism (Dwork & Roth, 2014), and the introduction of i.i.d. Gaussian noise for robustness to adversarial attacks in deep learning (Ying, 2019; Li et al., 2019) Numerous studies have investigated the impact of modeled noise, specifically Gaussian noise (Yu & Wong, 2009; Iam-on, 2020), on clustering tasks (Dave, 1993; Garc'ia-Escudero et al., 2008; Iam-on, 2020). They analyze the relation between the level of noise and the performance of clustering algorithms and show that small amount of noise may benefit centroid-based clustering methods (Iam-on, 2020).

Given the wide-ranging applications of coresets as a data reduction technique for clustering problems, it is thus natural to study them in the presence of noise. To measure the error due to a coreset $S$ with respect to $P$ (in the absence of noise), one considers the maximum ratio $\mathsf{Err}(S)$ between the cost difference $|\mathrm{cost}_z(S, C) - \mathrm{cost}_z(P, C)|$ and the original $\mathrm{cost}_z(P, C)$, over all center sets $C$ of size $k$ (Feldman & Langberg, 2011; Cohen-Addad et al., 2021). Previous research has primarily focused on analyzing the optimal tradeoff between the size $|S|$ of a coreset and the associated estimation error (Cohen-Addad et al., 2022; Cohen-Addad et al., 2022; Huang et al., 2023b). A natural idea is to extend this size-error tradeoff analysis for $\mathsf{Err}(S)$ to the setting of stochastic additive noise with known parameters. However, there are a few obstacles: 1) Due to the noise, the observed dataset $\widehat{P}$ may not be well-correlated with the true dataset $P$. Thus, even with the knowledge of the noise model and associated parameters, one can not expect to recover a coreset for $P$ from $\widehat{P}$. 2) Even if we let $S = \widehat{P}$, to estimate $\mathsf{Err}(\widehat{P})$, we need to upper bound $|\mathrm{cost}_z(\widehat{P}, C) - \mathrm{cost}_z(P, C)|$ for all possible center sets $C \subset \mathbb{R}^d$. Naively, one can get a bound on this in terms of the cumulative norms of noise over all datapoints. However, this approach is not tight (see Appendix B) and to more accurately quantify the effect of noise, we need to understand how noise may cancel. 3) Importantly, we find that the coreset error measure $\mathsf{Err}(\cdot)$ may be too strong to measure how good a center set we can obtain from $\widehat{P}$ in the presence of noise. This is because, due to noise, $\mathrm{cost}_z(\widehat{P}, C) - \mathrm{cost}_z(\widehat{P}, C')$ may have a different sign compared to $\mathrm{cost}_z(P, C) - \mathrm{cost}_z(P, C')$ for two center sets $C, C' \in \mathbb{R}$. Thus, roughly speaking, how good a center set we can obtain from $\widehat{P}$ is affected by the number of sign changes. Moreover, due to noise, $\mathrm{cost}_z(\widehat{P}, C) - \mathrm{cost}_z(\widehat{P}, C')$ may change significantly which makes $\mathsf{Err}(\widehat{P})$ large compared to the number of sign changes. Thus, a new measure is needed to account for this inconsistency and specifically, to quantify the quality of the optimal center set $\widehat{C}$ of $\widehat{P}$, i.e., how good $\widehat{C}$ is on $P$.

**Our contributions.** We study the problem of quantifying the quality of a coreset for clustering in the presence of noise. Motivated by the aforementioned applications, we consider the stochastic additive noise model in which each $D_j$ (as above) is parameterized by the *noise-level* $\theta$, and is only required to satisfy a certain bounded-moment condition (see Definition 2.1). This model includes distributions such as Gaussian and Laplace where $\theta$ is the variance. Conceptually, to address the issues emerging from the use of the $\mathsf{Err}(\cdot)$ discussed above, we propose a new notion $\mathsf{Err}_{\mathrm{AR}}(\cdot)$ of coreset performance (Definition 2.2), called *approximation-ratio*. Roughly, $\mathsf{Err}_{\mathrm{AR}}(\cdot)$ ensures that the optimal center set of $\widehat{P}$ is $(1 + \mathsf{Err}_{\mathrm{AR}}(\widehat{P}))$-approximately optimal for $P$ (Claim 2.3).

Technically, under mild assumptions on the dataset $P$ (Assumptions 1 and 2), we prove quantitative bounds on the coresets selected using this approximation-ratio measure as a function of the noise-

level $\theta$, the input parameters $n, k, \varepsilon$, and $\mathsf{OPT} := \min_{C \subset \mathbb{R}^d : |C| = k} \mathrm{cost}_z(P, C)$ (see Theorem 3.1). A consequence of Theorem 3.1 is that $\mathsf{Err}_{\mathrm{AR}}(\widehat{P}) = O(\frac{\theta k d}{\mathsf{OPT}})$, which implies that the optimal $k$-MEANS solution $\widehat{C}$ of $\widehat{P}$ is $(1 + O(\frac{\theta k d}{\mathsf{OPT}}))$-approximately optimal for $P$ (see Corollary 3.2). This approximation ratio is shown to be nearly tight in Theorem 3.3 and is better than the approximation ratio $1 + O(\frac{\theta n d}{\mathsf{OPT}})$ obtained by the coreset measure $\mathsf{Err}(\cdot)$. Additionally, we also provide bounds for $(k, z)$-CLUSTERINGin Section E. The key ingredient in the proof of Theorem 3.1 is an understanding of how the noise cancels (Claim 3.6). Finally, we show in Lemma 3.4 how Theorem 3.1 can be used to guide the selection of coreset size for any algorithm for a given noise level.

Empirically, in Section 4, we present estimates on $\mathsf{Err}$ and $\mathsf{Err}_{\mathrm{AR}}$ for coresets of different sizes for `Adult` and `Bank` datasets with additive Gaussian or Laplace noise. The empirical findings are consistent with the theoretical results and demonstrate the following: 1) A numerical separation exists between $\mathsf{Err}$ and $\mathsf{Err}_{\mathrm{AR}}$, in which the errors with respect to $\mathsf{Err}_{\mathrm{AR}}$ are smaller (see e.g., solid lines vs. dashed lines in Figure 1). 2) Both $\mathsf{Err}$ and $\mathsf{Err}_{\mathrm{AR}}$ initially decrease and then stabilize as the coreset size increases; this can be utilized to select an appropriate coreset size (see the changes in the columns of Table 1). 3) Variance of noise (or noise level) is the main parameter that determines both $\mathsf{Err}$ and $\mathsf{Err}_{\mathrm{AR}}$ (e.g., similar lines can be observed in all Figures 1a–1d).

## 2 MODEL

**Coreset for $(k, z)$-CLUSTERING.** Denote the collection of all subsets $C \subset \mathbb{R}^d$ of size $k \geq 1$ by $\mathcal{C}$. An $\varepsilon$-coreset for the $(k, z)$-CLUSTERING problem (which we define in Section 1) is a set $S \subset \mathbb{R}^d$ with a weight function $w : S \to \mathbb{R}_{\geq 0}$ such that $\mathrm{cost}_z(S, C) := \sum_{x \in S} w(x) \cdot d^z(x, C) \in (1 \pm \varepsilon) \cdot \mathrm{cost}_z(P, C)$ holds for every $C \in \mathcal{C}$. Define $\mathsf{Err}(S) := \sup_{C \in \mathcal{C}} \frac{|\mathrm{cost}_z(S, C) - \mathrm{cost}_z(P, C)|}{\mathrm{cost}_z(P, C)}$ to be the error measure of a coreset $S$ that bounds the maximum difference in the clustering cost of all center sets $C \in \mathcal{C}$ for $S$ versus that of $P$. This measure is well studied in the noise-free setting (Har-Peled & Mazumdar, 2004; Feldman & Langberg, 2011; Huang et al., 2019).

**Noise models.** Given a probability distribution $D$ on $\mathbb{R}$ with mean $\mu$ and variance $\sigma^2$, we say that $D$ satisfies the Bernstein condition if there exists some constant $b > 0$ such that for every integer $i \geq 3$, $\mathbb{E}_{X \sim D}\left[|X - \mu|^i\right] \leq \frac{1}{2}k!\sigma^2 b^{i-2}, i = 3, 4, \ldots$. This condition has been well studied in the literature (Bernstein, 1946; Bennett, 1962; Fan et al., 2012). It imposes an upper bound on each moment of $D$, which allows for control of tail behaviors. Multiple well-known distributions satisfy the Bernstein condition, including Gaussian distribution, Laplace distribution, sub-Gaussian distributions, sub-exponential distributions, and so on (Vershynin, 2018); see Appendix C.1 for a discussion. Next, we define the following noise model that we consider in this paper.

**Definition 2.1 (Noise model I)** *Let $\theta \in [0, 1]$ be a noise parameter and $D_1, \ldots, D_d$ be probability distributions on $\mathbb{R}$ with mean 0 and variance 1 that satisfy the Bernstein condition. Every point $\widehat{p}_i$ ($i \in [n]$) is i.i.d. drawn from the following distribution: 1) with probability $1 - \theta$, $\widehat{p}_i = p_i$; 2) with probability $\theta$, for every $j \in [d]$, $\widehat{p}_{i,j} = p_{i,j} + \xi_{p,j}$ where $\xi_{p,j}$ is drawn from $D_j$.*

Intuitively, we select a fraction $\theta$ of underlying points and add an independent noise to each feature $j \in [d]$ that is drawn from a certain distribution $D_j$. Note that as $\theta$ approaches 1, the estimate $\widehat{P}$ becomes increasingly noisy since the variance of $\xi_{p,j}$ is $\theta$ for each $p$ and $j$. Specifically, when $\theta = 0$, $\widehat{P}$ is noise-free and equal to $P$. Several applications mentioned in Section 1 apply this noise model, e.g., selecting $\theta = 1$ and each $D_j$ to be an appropriate Gaussian distribution (Dwork & Roth, 2014; Li et al., 2019).

Note that the variance of each $D_j$ can be fixed to any $t > 0$ since we can scale each point in the dataset by $\frac{1}{t}$. For the ease of measuring the impact of the variance of each $D_j$ on the performance of a coreset empirically, we also define the following model, called *noise model II*: For every $i \in [n]$ and $j \in [d]$, $\widehat{p}_{i,j} = p_{i,j} + \xi_{p,j}$ where $\xi_{p,j}$ is drawn from $D_j$, which is a probability distribution on $\mathbb{R}$ with mean 0, variance $\sigma^2$, and satisfies the Bernstein condition.

We study the performance of coreset for $(k, z)$-CLUSTERING under noise model I. Specifically, for a set $S$ obtained from $\widehat{P}$, we want to quantify how good a center set can one obtain for the (unknown) dataset $P$.

**Obstacles in using measure** $\mathsf{Err}(\cdot)$**.** A natural idea is to extend the prior analysis of the coreset measure $\mathsf{Err}(\cdot)$ to our noise model. However, this coreset measure could be too strong with noisy data. Take 1-MEANS as an example where there is only one center $c \in \mathbb{R}$, and consider $S = \widehat{P}$. We first observe that noise may lead to sign changes in cost, i.e., $\mathrm{cost}_2(\widehat{P}, c) - \mathrm{cost}_2(\widehat{P}, c')$ may have a different sign as $\mathrm{cost}_2(P, c) - \mathrm{cost}_2(P, c')$ for $c, c' \in \mathbb{R}$. Specifically, if there is no sign change from $P$ to $\widehat{P}$, the optimal center $c^\star \in \mathbb{R}^d$ of $P$ satisfies that $\mathrm{cost}_2(\widehat{P}, c^\star) = \min_{c \in \mathbb{R}^d} \mathrm{cost}(\widehat{P}, c)$. This implies that $c^\star$ is also an optimal center of $\widehat{P}$. Intuitively, how good a center set we can obtain from $\widehat{P}$ is affected by the number of sign changes. Also note that for any $c \in \mathbb{R}^d$, we have

$$\mathrm{cost}_2(\widehat{P}, c) - \mathrm{cost}_2(P, c) = \sum_{p \in P} \|\xi_p\|_2^2 + 2 \sum_{p \in P} \langle \xi_p, p - c \rangle,$$

which contains term $\sum_{p \in P} \|\xi_p\|_2^2$, that is the cumulative norms of $\xi_p$'s, and term $2 \sum_{p \in P} \langle \xi_p, p - c \rangle$, that is the cumulative inner-products between $\xi_p$ and $p - c$. We remark that the norm term $\sum_{p \in P} \|\xi_p\|_2^2$ can not cause sign changes, say for any two centers $c, c' \in \mathbb{R}$ with $\mathrm{cost}_2(P, c) < \mathrm{cost}_2(P, c')$, $\mathrm{cost}_2(P, c) + \sum_{p \in P} \|\xi_p\|_2^2 < \mathrm{cost}_2(P, c') + \sum_{p \in P} \|\xi_p\|_2^2$ still holds. In contrast, the inner-product term $2 \sum_{p \in P} \langle \xi_p, p - c \rangle$ varies for different centers $c$ and causes sign changes. We also note that the norm term can be much larger than the inner-product term and make the coreset measure $\mathsf{Err}(\widehat{P})$ large. Thus, $\mathsf{Err}(\cdot)$ can not quantify the number of sign changes, which motivates us to consider new measures to quantify the quality of the optimal center set $\widehat{C}$ of $\widehat{P}$ for $P$. See an illustration example in Appendix B.

**New coreset notion.** We propose a new notion of coreset (Definition 2.2) to bypass these obstacles. All proofs of its properties can be found in Appendix C.2. Given a dataset $P \subset \mathbb{R}^d$ of $n$ points, $\alpha \geq 1$, and a weighted point set $S \subset \mathbb{R}^d$ with weight $w : S \to \mathbb{R}_{\geq 0}$, we define $r_S(C) := \frac{\mathrm{cost}_z(S, C)}{\min_{C' \in \mathcal{C}} \mathrm{cost}_z(S, C')}$ to be the approximation ratio of $C$ on $S$ for every center set $C \in \mathcal{C}$. Moreover, we define $\mathcal{C}_\alpha(S) := \{C \in \mathcal{C} : r_S(C) \leq \alpha\}$ to be the collection of all $\alpha$-approximate solutions for $(k, z)$-CLUSTERING of $S$. Let $\mathsf{OPT}$ denote the optimal $(k, z)$-CLUSTERING cost of $P$.

**Definition 2.2 (Approximation-ratio coreset for** $(k, z)$**-CLUSTERING)** *Given a dataset $P \subset \mathbb{R}^d$ of $n$ points, $\varepsilon \in (0, 1)$, $\alpha \geq 1$, an $(\alpha, \varepsilon)$-approximation-ratio coreset for $(k, z)$-CLUSTERING is a weighted set $S \subset \mathbb{R}^d$ with weight $w : S \to \mathbb{R}_{\geq 0}$, such that $r_P(C) \leq (1 + \varepsilon) \cdot r_S(C)$ holds for every $C \in \mathcal{C}_\alpha(S)$. Define $\mathsf{Err}_{\mathrm{AR}}(S) := \sup_{C \in \mathcal{C}_\alpha(S)} \frac{r_P(C) - r_S(C)}{r_S(C)}$ to be the induced error measure.*

Compared to $\mathsf{Err}(S)$, our new measure $\mathsf{Err}_{\mathrm{AR}}(S)$ considers the ratio $r_S(C)$ for center sets instead of their cost $\mathrm{cost}_z(S, C)$. We note that the norm term $\sum_{p \in P} \|\xi_p\|_2^2$ does not cause a sign change in the ratio, i.e., $r_S(C) - r_S(C')$ has the same sign as $r_P(C) - r_P(C')$ for all $C, C' \in \mathcal{C}$. This property of the ratio aligns with that for sign changes in cost, which motivates this new measure $\mathsf{Err}_{\mathrm{AR}}(\cdot)$. By definition, we note that $S$ is an $(\alpha, \mathsf{Err}_{\mathrm{AR}}(S))$-approximation-ratio coreset of $P$ for any $\alpha \geq 1$, where $\mathsf{Err}_{\mathrm{AR}}(S))$ does not decrease as $\alpha$ increases. Also, a $\beta$-approximate center set $C \in \mathcal{C}_\alpha(S)$ of $S$ must be a $\beta(1 + \varepsilon)$-approximate center set of $P$, which allows us to find an approximately optimal center set of $P$ from $S$ (Claim 2.3). Parameter $\alpha$ controls the scale of the collection of restricted center sets we consider, which allows us to quantify the different impacts of noise on center sets $C$ with different levels of cost $\mathrm{cost}_z(S, C)$. In practice, We can set $\alpha = 1 + O(\varepsilon)$ since there exist fixed-parameter tractable (FPT) algorithms for $(k, z)$-CLUSTERING (Cohen-Addad et al., 2019).[1] We remark that our new notion is independent of noise and can be easily extended to measure the performance of coreset for other learning tasks like regression.

**Claim 2.3 (Approximation-ratio coreset to an approximate solution)** *Let $P \subset \mathbb{R}^d$ be a dataset of $n$ points and $\varepsilon, \varepsilon' \in (0, 1), \alpha \geq \alpha' \geq 1$. Suppose $S$ is an $(\alpha, \varepsilon)$-approximation-ratio coreset of $P$ for $(k, z)$-CLUSTERING and $C$ is an $\alpha'$-approximate solution of $S$ ($\alpha' \geq 1$). Then we have $C$ is an $\alpha'(1 + \varepsilon)$-approximate solution of $P$ for $(k, z)$-CLUSTERING.*

Both the coreset and approximation-ratio coreset exhibit the following composition property that proves advantageous in constructing a merge-and-reduce framework for coreset generation in dis-

---

[1] Here, an FPT algorithm finds a $(1 + \varepsilon)$-approximate solution for $(k, z)$-CLUSTERING in time $f(k, \varepsilon) \cdot \mathrm{poly}(n, d)$ where $f(k, \varepsilon)$ may not be a polynomial function.

tributed and streaming settings (Phillips, 2016; Braverman et al., 2016). For coreset, this composition property is known before; see e.g., (Har-Peled & Mazumdar, 2004, Observation 7.1) and (Feldman et al., 2011, Section 3.2).

**Claim 2.4 (Composition property)** *Let $P \subset \mathbb{R}^d$ be a dataset of $n$ points and $\varepsilon, \varepsilon' \in (0, 1), \alpha \geq 1$. We have 1) Suppose $S'$ is an $\varepsilon'$-coreset of $P$ for $(k, z)$-CLUSTERING and $S$ is an $\varepsilon$-coreset of $S'$ for $(k, z)$-CLUSTERING. Then $S$ is an $O(\varepsilon + \varepsilon')$-coreset of $P$. 2) Suppose $S'$ is an $(\alpha, \varepsilon')$-approximation-ratio coreset of $P$ for $(k, z)$-CLUSTERING and $S$ is an $\varepsilon$-coreset of $S'$ for $(k, z)$-CLUSTERING. Then $S$ is an $\left(\frac{\alpha}{1+O(\varepsilon)}, O(\varepsilon + \varepsilon')\right)$-approximation-ratio coreset of $P$.*

## 3 THEORETICAL RESULTS

We study the $k$-MEANS problem in the main body and defer the results to $(k, z)$-CLUSTERING to Appendix E. For simplicity, we use cost to replace $\text{cost}_2$ in the following context.

**Main result.** Theorem 3.1 provides nearly tight error bounds for two measures $\text{Err}(S)$ and $\text{Err}_{\text{AR}}(S)$ on a coreset $S$ obtained from $\widehat{P}$ under the noise model I. The theorem offers a separation between these two measures that can be verified empirically (Section 4), showing the effectiveness of our new measure $\text{Err}_{\text{AR}}(S)$ in the presence of noise. Due to certain technical difficulties, our results rely on mild assumptions on data (Assumptions 1 and 2), which we will explain later.

**Theorem 3.1 (Performance analysis for $k$-MEANS coreset under noise model I)** *Let $\varepsilon, \theta \in [0, 1]$ and $1 \leq \alpha \leq 1 + \frac{1}{k}$. Let $P \subset \mathbb{R}^d$ be an underlying dataset of size $n$ satisfying Assumptions 1 and 2 on $P$. Suppose $n \gg k^2$ is sufficiently large. Let $\widehat{P} \subset \mathbb{R}^d$ be an observed dataset drawn from $P$ under the noise model I with underlying parameter $\theta$. Let $S$ be an $\varepsilon$-coreset of $\widehat{P}$ for $k$-MEANS. With probability at least 0.8, $S$ is an $O(\varepsilon + \frac{\theta nd}{\text{OPT}} + \sqrt{\frac{\theta nd}{\text{OPT}}})$-coreset and an $\left(\frac{\alpha}{1+O(\varepsilon)}, O(\varepsilon + \frac{\theta kd}{\text{OPT}} + \frac{(1-\frac{1}{\alpha}+e^{-O(\sqrt{kd})})\theta nd}{\text{OPT}} + (1 - \frac{1}{\alpha}))\right)$-approximation-ratio coreset of $P$.*

We emphasize that the presented theorem is an analytical finding that does not need the information of the noise level $\theta$, rather than an algorithmic one, and can be used to derive estimates of the quality of a coreset produced by any algorithm in the noisy setting (see Lemma 3.4). Our result directly implies the following corollary.

**Corollary 3.2** $\text{Err}_{\text{AR}}(\widehat{P}) = O(\frac{\theta kd}{\text{OPT}})$ *when $\varepsilon = 0$, $\alpha = 1$ and $n \leq e^{O(\sqrt{kd})}k$.*

By Claim 2.3, this corollary implies that the optimal $k$-MEANS center set $\widehat{C}$ of $\widehat{P}$ is a $(1+O(\frac{\theta kd}{\text{OPT}}))$-approximate center set of $P$. This approximation ratio is tight by Theorem 3.3. In contrast, we know that $\text{Err}(\widehat{P}) = O(\frac{\theta nd}{\text{OPT}} + \sqrt{\frac{\theta nd}{\text{OPT}}})$-coreset of $P$, which only provides a guarantee that $\widehat{C}$ is a $(1 + O(\frac{\theta nd}{\text{OPT}} + \sqrt{\frac{\theta nd}{\text{OPT}}}))$-approximate center set of $P$. Since $\frac{\theta nd}{\text{OPT}} + \sqrt{\frac{\theta kd}{\text{OPT}}} \gg \frac{\theta kd}{\text{OPT}}$, we conclude that $\text{Err}_{\text{AR}}(\widehat{P})$ gets a refined estimate on the quality of $\widehat{C}$ than $\text{Err}(\widehat{P})$.

Another corollary of Theorem 3.1 is when $\varepsilon = 0$, $\alpha \geq 1 + \frac{k}{n} + e^{-O(\sqrt{kd})}$ and $\theta nd \geq \text{OPT}$, we obtain that $\text{Err}_{\text{AR}}(\widehat{P}) = O(\frac{(1-\frac{1}{\alpha})\theta nd}{\text{OPT}})$, which is also shown nearly tight by Theorem 3.3. This term increases to at most $O(\frac{\theta nd}{\text{OPT}})$ as $\alpha$ increases to $+\infty$, which implies that $\text{Err}(\cdot)$ is always an upper bound of $\text{Err}_{\text{AR}}(\cdot)$. This corollary indicates that the impacts of noise on the quality of center sets with different levels of cost are indeed different.

The success probability 0.8 can be fixed to any $\delta < 1$ by increasing the constant factor of error measures $\text{Err}(S)$ and $\text{Err}_{\text{AR}}(S)$ hidden in $O(\cdot)$. For approximation-ratio coreset, the constraint $\alpha \leq 1 + \frac{1}{k}$ is proposed to ensure that every $C \in \mathcal{C}_\alpha(\widehat{P})$ has the following structural property: for every $i \in [k]$, there must exist a center $c_i \in C \cap B(c_i^\star, \sqrt{k}a_i + \sqrt{kd})$ (Lemma D.5).

**Assumptions on data.** Now we discuss the assumptions on $P$ given in Theorem 3.1. By the composition property (Claim 2.4), we only need to give bounds for measures $\text{Err}(\widehat{P})$ and $\text{Err}_{\text{AR}}(\widehat{P})$,

which requires a quantitatively analysis on the values of $\text{cost}(\widehat{P}, C)$ over all center sets $C$. However, noise may change the assignments to centers in a center set $C$ from $P$ to $\widehat{P}$, i.e., the closest center of every pair $p$ and $\widehat{p}$ in $C$ may be different. The change in assignments raises difficulties in computing $\text{cost}(\widehat{P}, C)$. To bypass this, a natural idea is to introduce assumptions about data.

One common way is to assume an underlying generative distribution for data points, e.g., Gaussian mixture model (Feldman et al., 2011; Huang et al., 2021). For instance, we can assume each point $p \in P$ is i.i.d. drawn from a Gaussian mixture model $\sum_{i \in k} \frac{1}{k} N(\mu_i, 1)$ where all $\mu_i \in \mathbb{R}^d$ are far away from each other. Then the assignments between $P$ and $\widehat{P}$ are likely to be the same. However, this distributional assumption may be too strong since we only need its structural properties.

Now we provide our assumptions on data. Let $C^\star := \{c_1^\star, \ldots, c_k^\star\} \subset \mathbb{R}^d$ be the optimal $k$-MEANS center set of $P$ and $\text{OPT} := \text{cost}(P, C^\star)$ be the optimal $k$-MEANS cost. Let $P_1, \ldots, P_k \subset \mathbb{R}^d$ be a partition of $P$ where each $P_i$ contains $n_i$ points $p \in P$ whose closest center in $C^\star$ is $c_i^\star$ (breaking ties arbitrarily). For every $i \in [k]$, let $a_i := \max_{p \in P_i} d(p, c_i^\star)$ denote the radius of $P_i$. Then we have $P_i \subset B(c_i^\star, a_i)$ where $B(c_i^\star, a_i)$ is the ball centered at $c_i^\star$ of radius $a_i$. For every $i \in [k]$, we define $\text{OPT}_i := \text{cost}(P_i, c_i^\star)$ to be the cost of $P_i$ w.r.t. center set $C^\star$. Then $\text{OPT} = \sum_{i \in [k]} \text{OPT}_i$. We first give a balancedness assumption on $P$:

$$\frac{\max_{i \in [k]} \text{OPT}_i}{\min_{i \in [k]} \text{OPT}_i} = O(1) \text{ and } \frac{\max_{i \in [k]} n_i}{\min_{i \in [k]} n_i} = O(1). \tag{1}$$

This assumption requires both the contribution $\text{OPT}_i$ and the size $n_i$ of every cluster $P_i$ are at the same level, which has also been considered in the clustering literature (Bradley et al., 2000; Huang et al., 2023a). We also assume that $P_1, \ldots, P_k$ are well separated:

$$\forall 1 \le i < j \le k, d(c_i^\star, c_j^\star) \ge \Omega(\sqrt{k}) \cdot (a_i + a_j + \sqrt{d}). \tag{2}$$

The assumption of well-separated clusters implies that each cluster $P_i$ is far away from other clusters, whose idea has been widely considered in previous works (Han et al., 2012; Har-Peled & Rogge, 2021). Intuitively, Assumptions 1 and 2 ensure that every approximate solution $C \in \mathcal{C}$ is "close" to $C^\star$ and the assignments to centers of such $C$ between $P$ and $\widehat{P}$ are similar.

**Theorem 3.1 is nearly tight.** For 1-MEANS, we show the bounds in Theorem 3.1 are nearly tight for all datasets $P$ by the following theorem, whose proof can be found in Appendix D.2.

**Theorem 3.3 (Lower bound for 1-MEANS coreset under noise model I)** *Let $\varepsilon, \theta \in [0, 1]$, $\alpha \ge 1$ be constant. Let $P \subset \mathbb{R}^d$ be an underlying dataset of size $n \ge 1$ and $\widehat{P} \subset \mathbb{R}^d$ be an observed dataset drawn from $P$ under the noise model I with underlying parameter $\theta$. Suppose the variance of each $D_j$ ($j \in [d]$) is exactly $\theta$. With probability at least 0.8, $\widehat{P}$ is an $\Omega(\frac{\theta n d}{\text{OPT}})$-coreset and a $(1, \Omega(\frac{\theta d}{\text{OPT}}))$-approximation-ratio coreset of $P$ for 1-MEANS.*

*Moreover, assuming $1 - \frac{1}{\alpha} \gg \frac{1}{n}$, we have that $\widehat{P}$ is an $(\alpha, \Omega(\frac{(1-\frac{1}{\alpha})\theta n d}{\text{OPT}}))$-approximation-ratio coreset of $P$ for 1-MEANS, with probability at least 0.3.*

The bounds in this theorem can be easily extended to general $k$-MEANS: Consider a dataset $P$ contains $k$ copies $P_1, \ldots, P_k$ that are extremely well separated, e.g., $d(c_i^\star, c_j^\star) \to +\infty$ for every $1 \le i < j \le k$. In this case, each cluster $P_i$ is assigned to a unique center in $C$ with a finite cost $\text{cost}(P, C)$ – clusters are independent from each other. By calculations, we have that $\widehat{P}$ is an $\Omega(\frac{\theta n d}{\text{OPT}})$-coreset, a $(1, \Omega(\frac{\theta d}{\text{OPT}}))$-approximation-ratio coreset, and an $(\alpha, \Omega(\frac{(1-\frac{1}{\alpha})\theta n d}{\text{OPT}}))$ (when $1 - \frac{1}{\alpha} \gg \frac{k}{n}$) of $P$ for $k$-MEANS. Thus, Theorem 3.1 is nearly tight in the worst case.

**Guiding the selection of coreset size.** We discuss how to apply our theoretical results to guide the selection of coreset size for any algorithm in the presence of noise, whose relation is summarized in the following lemma. The correctness is simply guaranteed by Claim 2.4.

**Lemma 3.4 (Selection of coreset size in the presence of noise)** *Let $\mathcal{A}$ be a coreset algorithm that constructs an $\varepsilon$-coreset of $\widehat{P}$ for $(k, z)$-CLUSTERING of size $\mathcal{A}(\varepsilon)$ ($\varepsilon \ge 0$). Suppose that $\widehat{P}$ is an $\varepsilon'$-coreset (or $(\alpha, \varepsilon')$-approximation-ratio coreset) for $(k, z)$-CLUSTERING of $P$ ($\varepsilon' > 0$ and $\alpha \ge 1$). Then setting the coreset size to $\mathcal{A}(\varepsilon')$ achieves an $O(\varepsilon')$-coreset (or $(\frac{\alpha}{1+O(\varepsilon')}, O(\varepsilon'))$-approximation-ratio coreset) for $(k, z)$-CLUSTERING of $P$, and these errors can not be improved by increasing the coreset size.*

As a corollary together with Theorem 3.3, if we apply the coreset algorithm of Cohen-Addad et al. (2022) which constructs an $\varepsilon$-coreset for $k$-MEANS of size $\tilde{O}(k^{1.5}\varepsilon^{-2})$, the selection of the coreset size should be at most $\tilde{O}\left(\frac{k^{1.5}\cdot\mathsf{OPT}^2}{\theta^2 n^2 d^2}\right)$ for the coreset measure and at most $\tilde{O}\left(\frac{k^{1.5}\cdot\mathsf{OPT}^2}{\theta^2 d^2}\right)$ for the approximation-ratio measure. Our empirical results support this observation; see discussions in Section 4. In addition, we believe the choice of coreset strategy does not significantly impact the performance, due to the independent and directionally random nature of our noise models.

**Proof overview of Theorem 3.1.** By the composition property (Claim 2.4), the key is to study the relation of $\widehat{P}$ and $P$; summarized by the following lemma.

**Lemma 3.5 (Analysis of $\widehat{P}$)** *With probability at least 0.8, $\widehat{P}$ is an $O(\frac{\theta nd}{\mathsf{OPT}} + \sqrt{\frac{\theta nd}{\mathsf{OPT}}})$-coreset and an*
$\left(\alpha, O(\frac{\theta d}{\mathsf{OPT}} + \frac{(1-\frac{1}{\alpha}+e^{-O(\sqrt{kd})})\theta nd}{\mathsf{OPT}} + (1 - \frac{1}{\alpha}))\right)$*-approximation-ratio coreset of $P$ for $k$-MEANS.*

Theorem 3.1 is a direct corollary of Claim 2.4 and Lemma 3.5 by simply compositing errors. Hence, it remains to prove Lemma 3.5 and we give a proof sketch. We first prove for the case of $k = 1$, whose key idea is summarized by the following claim that analyzes the cancellation manner of noise.

**Claim 3.6 (Error analysis for 1-MEANS)** *For 1-MEANS, we have 1)* $\mathbb{E}_{\widehat{P}}\left[\sum_{p\in P}\|\xi_p\|_2^2\right] = \theta nd$
*and* $\mathrm{Var}_{\widehat{P}}\left[\sum_{p\in P}\|\xi_p\|_2^2\right] = O(\theta nd^2)$*; 2) With probability at least 0.95,* $\sup_{c\in\mathbb{R}^d}\frac{|\sum_{p\in P}\langle\xi_p, p-c\rangle|}{\mathrm{cost}(P,c)} = O\left(\sqrt{\frac{\theta d}{\mathsf{OPT}}}\right)$*; 3) The optimal solution of $\widehat{P}$ is* $\mu(\widehat{P}) := c^\star + \frac{1}{n}\sum_{p\in P}\xi_p$*, and* $\|\sum_{p\in P}\xi_p\|_2 = O(\sqrt{\theta nd})$ *with probability at least 0.95.*

This claim primarily relies on certain concentration properties of the terms $\sum_{p\in P}\|\xi_p\|_2^2$ and $\sum_{p\in P}\langle\xi_p, p - c^\star\rangle$, which is guaranteed by the Bernstein condition. Finally, we extend the analysis to general $k \geq 1$. The main idea is to show that under Assumptions 1 and 2, the assignments between $P$ and $\widehat{P}$ to any center set $C \in \mathcal{C}_\alpha(\widehat{P})$ are close enough (Lemma D.6).

Our proof has a byproduct that it is difficult to construct an $o(\frac{\theta d}{\mathsf{OPT}})$-coreset $S$ for $k$-MEANS of $P$ from $\widehat{P}$; see details in Appendix D.3.

## 4 EMPIRICAL RESULTS

This section presents empirical results that support our theoretical findings with different noise models. We investigate how the measures $\mathsf{Err}(S)$ and $\mathsf{Err}_{\mathrm{AR}}(S)$ changes as the size $|S|$ and the noise level varies, providing evidence for the validity of our theoretical analysis.

**Setup.** We consider the $k$-MEANS clustering problem on `Adult` (Kohavi & Becker, 1996) and `Bank` (Moro et al., 2014) dataset from UCI machine learning repository. The `Adult` dataset consists of 48842 data points and each data has 6 features. The `Bank` dataset consists of 41188 data points and each data has 10 features.[2] We set $k = 10$ for the $k$-MEANS clustering problem and study noise models I and II with both Gaussian and Laplace noise (Section 2) on `Adult` and `Bank` datasets. Under the noise model I, we perturb the original dataset $P$ with the noise levels $\theta$ ranging from $[0.02, 0.04, 0.08, 0.2, 0.4, 0.8]$ and get the perturbed dataset $\widehat{P}$. Under the noise model II, we do the same thing to get $\widehat{P}$ with the noise levels $\sigma^2$ also ranging from $[0.02, 0.04, 0.08, 0.2, 0.4, 0.8]$. Then, we compute coreset $S$ with different sizes ranging from 500 to 5000 on the perturbed dataset $\widehat{P}$.[3] Finally, we measure and compare the performances of the coresets $S$ with respect to the original dataset $P$ under different measurements (defined in the following paragraph). For every dataset, every noise model, and every noise level, we repeat the above procedures 20 times.

---

[2]We drop the categorical features and only keep the continuous features for clustering.

[3]We utilize the coreset algorithm presented in Feldman & Langberg (2011). It is worth noting that the choice of coreset algorithm can be arbitrary since our primary focus is on performance analysis rather than reducing the size of the coreset.

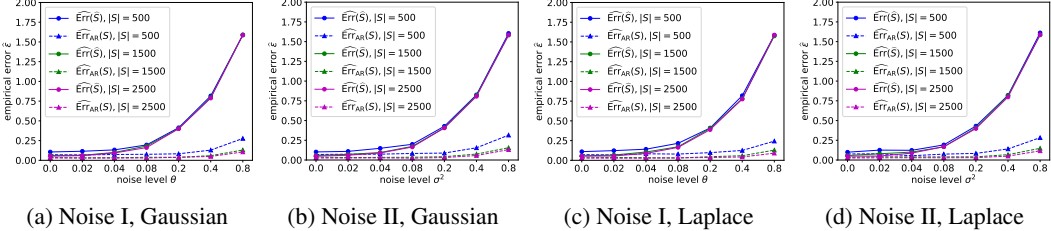

| (a) Noise I, Gaussian | (b) Noise II, Gaussian | (c) Noise I, Laplace | (d) Noise II, Laplace |

Figure 1: The empirical errors $\widehat{\mathrm{Err}}(S)$ and $\widehat{\mathrm{Err}}_{\mathrm{AR}}(S)$ versus the noise level ($\theta$ or $\sigma^2$) plot for coreset $S$ with different sizes. The solid lines in each figure represent the coreset measure $\widehat{\mathrm{Err}}(S)$, and the dashed lines represent the approximation-ratio measure $\widehat{\mathrm{Err}}_{\mathrm{AR}}(S)$. We show the results from `Bank` dataset, and Figures 1a to 1d denote the noise models I, II with Gaussian and Laplace noise respectively.

Table 1: Empirical errors $\widehat{\mathrm{Err}}(S)$ and $\widehat{\mathrm{Err}}_{\mathrm{AR}}(S)$ under different noise models, coreset sizes, and datasets (`Bank` and `Adult`). We fix the noise level $\theta = \sigma^2 = 0.2$. We repeat each setting 20 times, and provide the mean and the standard deviation (in the subscript) of the empirical errors.

(a) Results for `Bank` dataset.

| Size | Model I, Gaussian | | Model II, Gaussian | | Model I, Laplace | | Model II, Laplace | |
|---|---|---|---|---|---|---|---|---|
| | $\widehat{\mathrm{Err}}(S)$ | $\widehat{\mathrm{Err}}_{\mathrm{AR}}(S)$ | $\widehat{\mathrm{Err}}(S)$ | $\widehat{\mathrm{Err}}_{\mathrm{AR}}(S)$ | $\widehat{\mathrm{Err}}(S)$ | $\widehat{\mathrm{Err}}_{\mathrm{AR}}(S)$ | $\widehat{\mathrm{Err}}(S)$ | $\widehat{\mathrm{Err}}_{\mathrm{AR}}(S)$ |
| 500 | $0.413_{0.04}$ | $0.083_{0.02}$ | $0.428_{0.04}$ | $0.088_{0.02}$ | $0.410_{0.05}$ | $0.097_{0.03}$ | $0.430_{0.05}$ | $0.084_{0.02}$ |
| 1000 | $0.418_{0.04}$ | $0.057_{0.02}$ | $0.413_{0.03}$ | $0.057_{0.02}$ | $0.431_{0.03}$ | $0.064_{0.02}$ | $0.419_{0.04}$ | $0.054_{0.01}$ |
| 2000 | $0.394_{0.02}$ | $0.037_{0.01}$ | $0.404_{0.02}$ | $0.035_{0.01}$ | $0.395_{0.03}$ | $0.039_{0.01}$ | $0.407_{0.02}$ | $0.035_{0.01}$ |
| 3000 | $0.386_{0.02}$ | $0.033_{0.01}$ | $0.405_{0.02}$ | $0.033_{0.01}$ | $0.391_{0.02}$ | $0.027_{0.01}$ | $0.398_{0.02}$ | $0.031_{0.01}$ |
| 4000 | $0.380_{0.02}$ | $0.031_{0.01}$ | $0.393_{0.02}$ | $0.027_{0.01}$ | $0.385_{0.02}$ | $0.029_{0.02}$ | $0.391_{0.02}$ | $0.028_{0.01}$ |
| 5000 | $0.380_{0.02}$ | $0.029_{0.01}$ | $0.391_{0.01}$ | $0.025_{0.01}$ | $0.380_{0.02}$ | $0.027_{0.01}$ | $0.391_{0.02}$ | $0.026_{0.01}$ |

(b) Results for `Adult` dataset.

| Size | Model I, Gaussian | | Model II, Gaussian | | Model I, Laplace | | Model II, Laplace | |
|---|---|---|---|---|---|---|---|---|
| | $\widehat{\mathrm{Err}}(S)$ | $\widehat{\mathrm{Err}}_{\mathrm{AR}}(S)$ | $\widehat{\mathrm{Err}}(S)$ | $\widehat{\mathrm{Err}}_{\mathrm{AR}}(S)$ | $\widehat{\mathrm{Err}}(S)$ | $\widehat{\mathrm{Err}}_{\mathrm{AR}}(S)$ | $\widehat{\mathrm{Err}}(S)$ | $\widehat{\mathrm{Err}}_{\mathrm{AR}}(S)$ |
| 500 | $0.370_{0.05}$ | $0.079_{0.02}$ | $0.359_{0.05}$ | $0.101_{0.03}$ | $0.361_{0.04}$ | $0.082_{0.03}$ | $0.355_{0.04}$ | $0.101_{0.03}$ |
| 1000 | $0.345_{0.04}$ | $0.036_{0.01}$ | $0.351_{0.04}$ | $0.063_{0.02}$ | $0.334_{0.04}$ | $0.044_{0.02}$ | $0.351_{0.03}$ | $0.055_{0.02}$ |
| 2000 | $0.352_{0.03}$ | $0.023_{0.01}$ | $0.342_{0.02}$ | $0.043_{0.02}$ | $0.343_{0.03}$ | $0.028_{0.01}$ | $0.344_{0.03}$ | $0.035_{0.01}$ |
| 3000 | $0.343_{0.03}$ | $0.016_{0.01}$ | $0.341_{0.02}$ | $0.030_{0.01}$ | $0.345_{0.02}$ | $0.019_{0.01}$ | $0.340_{0.02}$ | $0.023_{0.01}$ |
| 4000 | $0.340_{0.02}$ | $0.012_{0.01}$ | $0.347_{0.02}$ | $0.023_{0.01}$ | $0.343_{0.02}$ | $0.011_{0.00}$ | $0.343_{0.02}$ | $0.025_{0.01}$ |
| 5000 | $0.342_{0.02}$ | $0.010_{0.01}$ | $0.341_{0.02}$ | $0.025_{0.01}$ | $0.342_{0.02}$ | $0.012_{0.01}$ | $0.345_{0.02}$ | $0.018_{0.01}$ |

**Performance measurements.** Given a weighted set of data $S \subseteq \widehat{P}$, we denote two measures $\widehat{\mathrm{Err}}(S)$ and $\widehat{\mathrm{Err}}_{\mathrm{AR}}(S)$ to estimate the performance guarantees under the coreset and approximation-ratio coreset notions respectively. We randomly sample 500 $k$-center sets $C_1, \ldots, C_{500}$ and set

$$\widehat{\mathrm{Err}}(S) := \max_{1 \leq i \leq 500} \frac{|\mathrm{cost}(S, C_i) - \mathrm{cost}(P, C_i)|}{\mathrm{cost}(P, C_i)}$$

as a proxy of the $\mathrm{Err}(S)$ parameter in the coreset definition. To compute $\widehat{\mathrm{Err}}_{\mathrm{AR}}(S)$, we first apply KMeans++ algorithm (Arthur & Vassilvitskii, 2007) implemented by scikit-learn (Pedregosa et al., 2011) to find a solution $C^\star$ on the original dataset $P$ and $C_S$ on the coreset $S$. Then, we define

$$\widehat{\mathrm{Err}}_{\mathrm{AR}}(S) := \frac{\mathrm{cost}(P, C_S)}{\mathrm{cost}(P, C^\star)} - 1$$

as a proxy of the $\mathrm{Err}_{\mathrm{AR}}(S)$ parameter when $\alpha = 1$ in the approximation-ratio coreset definition. Both heuristics are commonly used in the coreset literature (Huang et al., 2019; Baker et al., 2020; Huang et al., 2022a).

**Performance analysis.** Figure 1 and Table 1 summarize our empirical results. Figure 1 shows the relation between the empirical errors $\widehat{\mathrm{Err}}(S)$, $\widehat{\mathrm{Err}}_{\mathrm{AR}}(S)$ and the noise level ($\theta$ or $\sigma^2$) for coreset $S$ with different sizes on `Bank` dataset. Table 1 summarizes empirical errors at a fixed noise level

$\theta = \sigma^2 = 0.2$ on both `Bank` and `Adult` datasets. Then we discuss our empirical findings from Figure 1 and Table 1.

*Gap between* $\widehat{\mathrm{Err}}(S)$ *and* $\widehat{\mathrm{Err}}_{\mathrm{AR}}(S)$. First from Figure 1 and Table 1, we observe that there is a significant gap between $\widehat{\mathrm{Err}}(S)$ and $\widehat{\mathrm{Err}}_{\mathrm{AR}}(S)$, under different noise models with different noise levels $\theta$ and different sizes $|S|$. In particular, for a fixed noise level such as $\theta = 0.2$ in Table 1, the approximation-ratio measure $\widehat{\mathrm{Err}}_{\mathrm{AR}}(S)$ exhibits larger variablility compared to vanilla coreset measure $\widehat{\mathrm{Err}}(S)$ for different coreset sizes (see the changes in the columns of Table 1). It indicates that $\widehat{\mathrm{Err}}_{\mathrm{AR}}(S)$ measures the quality of a center set that we can obtain from $S$ on $P$ better under the noisy data setting. These observations match our theoretical findings in Theorem 3.1.

*The effect of different coreset sizes.* As presented in Table 1, both $\widehat{\mathrm{Err}}(S)$ and $\widehat{\mathrm{Err}}_{\mathrm{AR}}(S)$ initially decrease and then stabilize as the coreset size $|S|$ increases. For example, in the case of the `Bank` dataset under the noise model II with Gaussian noise, $\widehat{\mathrm{Err}}(S)$ decreases from 0.428 to 0.404 as $|S|$ increases from 500 to 2000, and then remains around 0.4 as $|S|$ continues to increase to 5000. This observation provides empirical evidence for our theoretical findings that the performance of the coreset is mainly affected by the noise when the size $|S|$ is sufficiently large. Therefore, this observation can assist in guiding the selection of appropriate coreset sizes for noisy datasets.

*The effect of different noise/variance levels.* When increasing $\theta$ under the noise model I or increasing $\sigma^2$ under the noise model II, both $\widehat{\mathrm{Err}}(S)$ and $\widehat{\mathrm{Err}}_{\mathrm{AR}}(S)$ increase no matter the type of the noise (see Figures 1a–1d). We also observe that these lines are similar across all Figures 1a–1d. This similarity may arise from the same variance of noise $\xi_{p,j}$ for both noise models I and II with Gaussian and Laplace noises when $\theta = \sigma^2$,[4] indicating that variance of noise (or noise level) is the main parameter that determines both $\mathrm{Err}$ and $\mathrm{Err}_{\mathrm{AR}}$. Additionally, we observe that $\widehat{\mathrm{Err}}_{\mathrm{AR}}(S)$ exhibits a relatively small increase even when the noise level $\theta$ is significantly larger (40 times larger in our experiments, from 0.02 to 0.8), indicating the robustness of the optimal solution $C_S$ in the presence of noise.

*Additional empirical results.* We also conduct experiments for different choices of $k$, examine whether Assumptions 1 and 2 hold on both `Bank` and `Adult` datasets, and report the changes of the total sensitivity under the noise model I. See Section F for these empirical results.

## 5 CONCLUSION, LIMITATIONS, AND FUTURE WORK

In this work, we investigate the efficacy of coresets for clustering in the presence of noise. We show that the prior measure to assess the quality of a coreset may be too strong for noisy data and propose a new performance measure. We provide a quantitative analysis of both measures under mild assumptions on data. The theoretical results verify the effectiveness of the new notion in being able to better capture the impact of noise on different center sets, compared to the prior measure. The empirical observations support the theoretical results and show that the variance of noise is the main factor that determines the quality of a coreset. This work provides guidance for selecting the coreset size for any algorithm in the presence of noise, which contributes to the broader field of machine learning in the context of noisy data.

One limitation is that Theorem 3.1 relies on (mild) assumptions on data and it is an interesting direction to investigate to what extent our results hold without any assumptions. Such a result would require one to handle multiple possible ways to assign points from $P$ to centers in center sets and to deal with the inconsistency in assignments between $P$ and $\widehat{P}$. Another future work is to expand our analysis to encompass additional noise models, for instance, which may not be independent. It would also be interesting to study the influence of adversarial noise. Another area to explore is the influence of noise on coresets in other learning tasks such as regression and classification. It would be also intriguing to examine connections between coresets and other notions of robustness for clustering.

---

[4]Recall that under the noise model I, the variance of $\xi_{p,j}$ is $\theta$ for every point $p \in P$ and feature $j \in [d]$ when $D_j$ is a Gaussian distribution or a Laplace distribution with variance 1. In contrast, the variance of $\xi_{p,j}$ under the noise model II is $\sigma^2$.

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

CONTENTS

## A ADDITIONAL RELATED WORK

**Coresets for clustering.** There exists a substantial body of work focused on constructing coresets for $(k, z)$-CLUSTERING in various metric spaces, including Euclidean metrics, doubling metrics, shortest path metrics on graphs, and general discrete metrics (Har-Peled & Mazumdar, 2004; Feldman & Langberg, 2011; Feldman et al., 2013; Braverman et al., 2016; Huang et al., 2018; Cohen-Addad et al., 2021; 2022; Huang et al., 2023b). Researchers have also studied another notion called weak coresets for clustering problems (Munteanu & Schwiegelshohn, 2018; Cohen-Addad et al., 2021; Danos, 2021; Huang et al., 2023a), that only restrict to $O(1)$-approximate solutions on coresets. However, it is worth noting that there is no significant advantage in terms of coreset size when using weak coresets over regular ones.

**Clustering with noise.** The problem of clustering in the presence of noise has been extensively studied for several decades. Within the existing literature, two main settings have emerged. The first setting considers the noise in the data as being generated according to a specific distribution (Dave, 1993; Garc'ia-Escudero et al., 2008). The second assumes that the noise is generated adversarially, albeit with a limited number of adversaries (Balcan et al., 2008; Ben-David & Haghtalab, 2014; Balcan & Liang, 2016; Kushagra et al., 2016; 2017). In our work, we primarily focus on the setting that assumes the noise is generated from a known distribution.

The study of noisy clustering can be broadly categorized into two directions. The first direction involves developing methods to measure the robustness of clustering algorithms in the presence of noise (Dave, 1991; Hampel, 1971; Hennig, 2008). These methods highlight the vulnerability of traditional algorithms with a fixed number of clusters (Ackerman et al., 2013; Hennig, 2008). The second direction focuses on designing robust clustering algorithms that can effectively handle noisy data (Cuesta-Albertos et al., 1997; Dave, 1993; Ben-David & Haghtalab, 2014; Kushagra et al., 2017). Within this direction, there exists a related problem called robust clustering, where the clustering process allows for the omission of certain "outlier" points (Charikar et al., 2001; Chen, 2008; Gupta et al., 2017; Schelling & Plant, 2018; Friggstad et al., 2019; Roizman et al., 2019; Statman et al., 2020). In our work, we are primarily aligned with the first direction, as our main focus is on analysis, and the coreset guarantees we provide can be viewed as a measure of robustness.

## B OBSTACLES STATED IN SECTION 2

We provide more details on the three obstacles as stated in Section 2. Consider a simple example of 1-MEANS in $\mathbb{R}$, i.e., $k = d = 1$. Let $P \subset \mathbb{R}$ consist of a collection $P_-$ of $\frac{n}{2}$ points at -1 and a collection $P_+$ of $\frac{n}{2}$ points at 1. The optimal center is $c^\star = 0$ and $\mathsf{OPT} = n$. Obviously, $S = \{-1, 1\}$ with $w(-1) = w(1) = \frac{n}{2}$ is a 0-coreset of $P$ for 1-MEANS. Let $\widehat{P} \subset \mathbb{R}$ be an observed dataset drawn from $P$ under the noise model II with each $D_j = N(0, 1)$. Additionally, note that $\mathrm{cost}(P, -1) = \mathrm{cost}(P, 1) = 2n$, i.e., $c = -1$ and $c = 1$ are two 2-approximate solutions of $P$.

**Obstacle 1: Prior coreset guarantees do not work.** By construction, the data distribution of $\widehat{P}$ is close to the following: each noisy point $\widehat{p} \in \widehat{P}$ is i.i.d. drawn from a certain Gaussian mixture model $\frac{1}{2}N(-1, 1) + \frac{1}{2}N(1, 1)$. Then

$$\mathbb{E}_{\widehat{P}}[\mathrm{cost}(\widehat{P}, 0)] = n \cdot \mathbb{E}_{X \sim N(0,1)}[(1 + X)^2] = 2n.$$

Recall that $\mathrm{cost}(S, 0) = \mathrm{cost}(P, 0) = n$. Thus, $S$ is even not a 0.9-coreset of $\widehat{P}$ for 1-MEDIAN instead of a 0-coreset.

In conclusion, introducing noise adds additional difficulties to the previous analysis of coreset in the noiseless setting. Even the clustering cost of the optimal solution of $P$ can have a significant change. One can quickly check that if we add a much larger noise, e.g., $N(0, 10)$ to each point, $\mathrm{cost}(\widehat{P}, 0)$ will grow very fast. Intuitively, we expect to obtain a quantitative analysis of the impact of noise, which should depend on noise parameters.

**Obstacle 2: How to quantify the impact of noise.** To quantify the effect of noise, a direct idea is to apply the well-studied coreset measure, i.e., to compute $\max_{c \in \mathbb{R}} \frac{|\mathrm{cost}(\widehat{P},c) - \mathrm{cost}(P,c)|}{\mathrm{cost}(P,c)}$. Let $\xi_p = \widehat{p} - p$ be the noise vector of $p$. By simple computation, we have

$$\mathrm{cost}(\widehat{P}, c) - \mathrm{cost}(P, c) = \sum_{p \in P} \|\xi_p\|_2^2 + 2 \sum_{p \in P} \langle \xi_p, p - c \rangle. \tag{3}$$

It is not hard to verify that the first term $\sum_{p \in P} \|\xi_p\|_2^2$ concentrates on $n$. Then the main issue is to bound the second term $2 \sum_{p \in P} \langle \xi_p, p - c \rangle$. A naive approach is to bound each inner product $\langle \xi_p, p - c \rangle$ separately, i.e.,

$$2 \sum_{p \in P} \langle \xi_p, p - c \rangle \leq \sum_{p \in P} \|\xi_p\|_2^2 + \|p - c\|_2^2 = \mathrm{cost}(P, c) + \sum_{p \in P} \|\xi_p\|_2^2 \approx \mathrm{cost}(P, c) + n,$$

which leads to the following upper bound

$$\max_{c \in \mathbb{R}} \frac{|\mathrm{cost}(\widehat{P}, c) - \mathrm{cost}(P, c)|}{\mathrm{cost}(P, c)} \approx \max_{c \in \mathbb{R}} \frac{\mathrm{cost}(P, c) + 2n}{\mathrm{cost}(P, c)} = 3.$$

This computation only shows that $\widehat{P}$ is a 3-coreset of $P$.

Consider the general case that $P \subset \mathbb{R}^d$ under the noise model I, we note that this idea leads to a bound $\mathrm{cost}(P, c) + \sum_{p \in P} \|\xi_p\|_2 \approx \mathrm{cost}(P, C) + \theta nd$. Then we can only obtain an upper bound $\mathrm{Err}(\widehat{P}) \leq 1 + \frac{2\theta nd}{\mathrm{OPT}}$, which is far from tight since noise may cancel with each other. Hence, the understanding of the cancellation manner of noise is important for quantifying its impact. By Claim 3.6, we will see a tight bound of the term $2 \sum_{p \in P} \langle \xi_p, p - c \rangle$, leading to $\mathrm{Err}(\widehat{P}) \leq O(\frac{\theta nd}{\mathrm{OPT}})$.

**Obstacle 3: Prior notion of coreset is too strong.** Let $\widehat{c}$ denote the optimal center of $\widehat{P}$. By Equation 3, we observe that the change $\mathrm{cost}(\widehat{P}, c) - \mathrm{cost}(P, c)$ can be large up to $\Omega(n)$, indicating that $\mathrm{Err}(\widehat{P}) \geq 1$. This measure is large and only implies that $\widehat{c}$ is a 2-approximate solution of $P$. However, since the cost difference on the right-side of Equation 3 contains a norm term $\sum_{p \in P} \|\xi_p\|_2^2$ for all centers $c \in C$, we claim that the measure $\mathrm{Err}(\widehat{P})$ may not be sufficient to capture the quality of $\widehat{c}$ on $P$: $r_P(\widehat{c}) = \frac{\mathrm{cost}(P, \widehat{c})}{\mathrm{OPT}}$. Consider the case that $\mathrm{cost}(\widehat{P}, c) - \mathrm{cost}(P, c) \approx n$ for all centers $c \in \mathbb{R}$, which actually holds by Claim 3.6. Given an arbitrary center $c \in \mathbb{R}$ with $\mathrm{cost}(P, c) - \mathrm{OPT} \ll n$, we have

$$r_{\widehat{P}}(c) = \frac{\mathrm{cost}(\widehat{P}, c)}{\mathrm{cost}(\widehat{P}, \widehat{c})} = \frac{\mathrm{cost}(\widehat{P}, c)}{\mathrm{cost}(P, \widehat{c}) + n} \leq \frac{\mathrm{cost}(\widehat{P}, c)}{\mathrm{cost}(P, c^\star) + n} < \frac{\mathrm{cost}(\widehat{P}, c)}{\mathrm{cost}(P, c)} \leq 1 + \mathrm{Err}(\widehat{P}),$$

which implies that the relative gap of the ratio $\frac{r_{\widehat{P}}(\widehat{c})}{r_P(\widehat{c})}$ can be more stable under noise compared to the cost $\frac{\mathrm{cost}(\widehat{P}, c)}{\mathrm{cost}(P, c)}$. Specifically, by simple calculation, we know that $\widehat{c} = \frac{1}{n} \sum_{p \in P} \xi_p$, which ensures that $|\widehat{c} - c^\star| = O(\sqrt{\frac{1}{n}})$. Then $\mathrm{cost}(P, \widehat{c}) = n + O(1)$ which implies that $\widehat{c}$ is a $(1 + O(\frac{1}{n}))$-approximate solution of $P$.

By this observation, the coreset measure $\mathrm{Err}(\widehat{P})$ is too strong for evaluating the quality of center sets in the presence of noise, specifically for $\widehat{c}$. This motivates us to propose the new notion of coreset (Definition 2.2) that considers ratio instead of cost. Interestingly, our theoretical results in Theorem 3.1 demonstrate a quantitative distinction between the measures of coreset and approximation-ratio coreset, which verifies the weakness of coreset measure in the presence of noise.

## C  MISSING DETAILS IN SECTION 2

### C.1  DISCUSSION ON THE BERNSTEIN CONDITION

We provide more information on the Bernstein condition. Specifically, we show that it is very general and covers sub-gaussian distributions and sub-exponential distributions.

We call $D$ a sub-gaussian distribution if there exists some constant $K > 0$ such that for any $t > 0$,

$$\Pr_{X \sim D}[|X| \geq t] \leq 2e^{-\frac{t^2}{K^2}};$$

and call $D$ a sub-exponential distribution if there exists some constant $K > 0$ such that for any $t > 0$,

$$\Pr_{X \sim D}[|X| \geq t] \leq 2e^{-t/K}.$$

Note that a Gaussian distribution is sub-gaussian and a Laplace distribution is sub-exponential. Moreover, we have the following well-known fact.

**Lemma C.1 (Moment bounds (Vershynin, 2018, Sections 2.5 and 2.7))** *If $D$ is a sub-gaussian distribution, there exists some constant $K > 0$ such that for all integers $i \geq 1$,*

$$\mathbb{E}_{X \sim D}[|X|^i]^{1/i} \leq K\sqrt{i}.$$

*If $D$ is a sub-exponential distribution, there exists some constant $K > 0$ such that for all integers $i \geq 1$,*

$$\mathbb{E}_{X \sim D}[|X|^i]^{1/i} \leq Ki.$$

This lemma actually implies that both sub-gaussian and sub-exponential distributions satisfy the Bernstein condition by combining with Stirling's approximation $i! \approx \sqrt{2\pi i} \cdot (\frac{i}{e})^i$.

Note that distributions with heavy tails may not satisfy Bernstein condition, e.g., Pareto distribution, log-normal distribution, and student's $t$-distribution.

## C.2   MISSING PROOFS IN SECTION 2

**Proof of Claim 2.3: Approximation-ratio coreset to an approximate solution.**  By Definition 2.2, we have $\alpha' \geq \frac{\text{cost}_z(S,C)}{\min_{C' \in \mathcal{C}} \text{cost}_z(S,C')} \geq \frac{\text{cost}_z(P,C)}{(1+\varepsilon) \cdot \text{OPT}}$, which implies that $\text{cost}_z(P,C) \leq \alpha'(1+\varepsilon) \cdot \text{OPT}$.

**Proof of Claim 2.4: Composition property.**  Suppose $S'$ is an $\varepsilon'$-coreset of $P$. For every center set $C \in \mathcal{C}$, we have

$$
\begin{aligned}
\text{cost}_z(S,C) &\in & (1 \pm \varepsilon) \cdot \text{cost}_z(S',C) && \text{(by assumption of } S) \\
&\in & (1 \pm \varepsilon) \cdot (1 \pm \varepsilon') \cdot \text{cost}_z(P,C) && \text{(by assumption of } S') \\
&\in & (1 \pm 2\varepsilon \pm 2\varepsilon') \cdot \text{cost}_z(P,C),
\end{aligned}
$$

which implies that $S'$ is a $2(\varepsilon + \varepsilon')$-coreset of $P$.

Next, suppose $S'$ is an $(\alpha, \varepsilon')$-approximation-ratio coreset of $P$. For every $C \in \mathcal{C}$ such that $C$ is an $\alpha'$-approximate solution of $S$ with $\alpha' \leq \frac{\alpha}{1+O(\varepsilon)}$, we have

$$
\begin{aligned}
\frac{\text{cost}_z(S',C)}{\min_{C' \in \mathcal{C}} \text{cost}_z(S',C')} &\leq & (1 + O(\varepsilon)) \cdot \frac{\text{cost}_z(S,C)}{\min_{C' \in \mathcal{C}} \text{cost}_z(S,C')} && \text{(by assumption of } S) \\
&\leq & (1 + O(\varepsilon)) \cdot \alpha' && \text{(by defn. of } C) \\
&\leq & \alpha,
\end{aligned}
$$

which implies that $C$ is at most an $\alpha$-approximate solution of $S'$. Then we have

$$
\begin{aligned}
r_S(C) &\geq & \frac{\text{cost}_z(S',C)}{(1 + O(\varepsilon)) \cdot \min_{C' \in \mathcal{C}} \text{cost}_z(S',C')} && \text{(by assumption of } S) \\
&\geq & \frac{r_P(C)}{(1 + O(\varepsilon)) \cdot (1 + \varepsilon')} && \text{(by assumption of } S') \\
&\geq & \frac{r_P(C)}{(1 + O(\varepsilon + \varepsilon'))},
\end{aligned}
$$

which completes the proof.

# D    MISSING DETAILS IN SECTION 3

## D.1    PROOF OF LEMMA 3.5: ANALYSIS OF $\widehat{P}$

In this section, we prove Lemma 3.5, which is the main lemma for Theorem 3.1. The proof is divided into two parts: the proof for the coreset guarantee, and the proof for the approximation-ratio coreset guarantee. For each part, we first prove for the case of $k = 1$ and then show how to extend to general $k \geq 1$.

**1. Proof for** $\mathrm{Err}(\cdot)$ **for 1-MEANS.**    We first prove Claim 3.6 by Claim D.1 and Lemma D.2. Note that the third property of Claim 3.6 is a direct corollary.

**Claim D.1 (Statistics of cost difference)** *For any center $c \in \mathbb{R}^d$, we have*

$$\mathrm{cost}(\widehat{P}, c) - \mathrm{cost}(P, c) = \sum_{p \in P} \|\xi_p\|_2^2 + 2 \sum_{p \in P} \langle \xi_p, p - c \rangle.$$

*Moreover, we have*

- $\mathbb{E}_{\widehat{P}} \left[ \sum_{p \in P} \|\xi_p\|_2^2 \right] = \theta n d$ *and* $\mathrm{Var}_{\widehat{P}} \left[ \sum_{p \in P} \|\xi_p\|_2^2 \right] = O(\theta n d^2)$;

- $\mathbb{E}_{\widehat{P}} \left[ \sum_{p \in P} \langle \xi_p, p - c \rangle \right] = 0$ *and* $\mathrm{Var}_{\widehat{P}} \left[ \sum_{p \in P} \langle \xi_p, p - c \rangle \right] = \theta \cdot \mathrm{cost}(P, c)$.

**Proof:**    We have $\mathrm{cost}(\widehat{P}, c) - \mathrm{cost}(P, c) = \sum_{p \in P} d^2(\widehat{p}, c) - d^2(p, c) = \sum_{p \in P} \|\xi_p\|_2^2 + 2\langle \xi_p, p - c \rangle$. For the first error term $\sum_{p \in P} \|\xi_p\|_2^2$, we have $\mathbb{E}_{\widehat{P}} \left[ \sum_{p \in P} \|\xi_p\|_2^2 \right] = \theta n \cdot \mathbb{E}_{x \sim N(0, I_d)} \left[ \|x\|_2^2 \right] = \theta n d$, and

$$
\begin{aligned}
& \mathrm{Var}_{\widehat{P}} \left[ \sum_{p \in P} \|\xi_p\|_2^2 \right] \\
= {} & n \cdot \mathrm{Var}_{\xi_p} \left[ \|\xi_p\|_2^2 \right] \\
= {} & n \cdot \left( \mathbb{E}_{\xi_p} \left[ \|\xi_p\|_2^4 \right] - \mathbb{E}_{\xi_p} \left[ \|\xi_p\|_2^2 \right]^2 \right) \\
= {} & n \cdot \left( \theta \cdot \mathrm{Var}_{T \sim \chi^2(d)} [T] + \theta \cdot \mathbb{E}_{x \sim N(0, I_d)} \left[ \|x\|_2^2 \right]^2 - \theta^2 \cdot \mathbb{E}_{x \sim N(0, I_d)} \left[ \|x\|_2^2 \right]^2 \right) \\
= {} & \theta n \cdot (2d + d^2 - \theta d^2) \\
\leq {} & 3\theta n d^2.
\end{aligned}
$$

where $\chi^2(d)$ represents the chi-square distribution with $d$ degrees of freedom, whose variance is known to be $2d$ Miller (2017).

For the second error term $\sum_{p \in P} \langle \xi_p, p - c \rangle$, its expectation is obvious 0 and we have

$$
\begin{aligned}
\mathrm{Var}_{\widehat{P}} \left[ \sum_{p \in P} \langle \xi_p, p - c \rangle \right] &= \sum_{p \in P} \mathrm{Var}_{\xi_p} \left[ \langle \xi_p, p - c \rangle \right] \\
&= \sum_{p \in P} \mathbb{E}_{\xi_p} \left[ \langle \xi_p, p - c \rangle^2 \right] \\
&= \theta \cdot \sum_{p \in P} \|p - c\|_2^2 \\
&= \theta \cdot \mathrm{cost}(P, c).
\end{aligned}
$$

$\square$

By Claim 3.6, it suffices to prove that

$$\sup_{c \in \mathbb{R}^d} \frac{\left| \sum_{p \in P} \|\xi_p\|_2^2 + 2 \sum_{p \in P} \langle \xi_p, p - c \rangle \right|}{\mathrm{cost}(P, c)} \leq O\left( \frac{\theta n d}{\mathsf{OPT}} + \sqrt{\frac{\theta d}{\mathsf{OPT}}} \right). \tag{4}$$

By Claim 3.6 and Chebyshev's inequality, we directly have that $\sum_{p \in P} \|\xi_p\|_2^2 \leq O(\theta n d)$ happens with probability at least 0.95. For the second error term $2 \sum_{p \in P} \langle \xi_p, p - c \rangle$, we provide an upper bound by the following lemma, which strengthens the second property of Claim 3.6.

**Lemma D.2** *With probability at least* $1 - 0.05\delta$ *for* $0 < \delta \leq 1$, *the following holds*

$$\sup_{c \in \mathbb{R}^d} \frac{|\sum_{p \in P} \langle \xi_p, p - c \rangle|}{\mathrm{cost}(P, c)} = O\left( \sqrt{\frac{\theta d}{\delta \mathsf{OPT}}} \right).$$

**Proof:** Let $X = |\{p \in P : \xi_p \neq 0\}|$ be a random variable. When $\theta n \leq 0.01\delta$, we have
$$\Pr[X = 0] = (1 - \theta)^n \geq 1 - 0.02\delta.$$
Conditioned on the event that $X = 0$, we have

$$\sup_{c \in \mathbb{R}^d} \frac{|\sum_{p \in P} \langle \xi_p, p - c \rangle|}{\mathrm{cost}(P, c)} = 0,$$

which completes the proof.

In the following, we analyze the case that $\theta n > 0.01\delta$. Let $E$ be the event that $\|\sum_{p \in P} \xi_p\|_2 \leq O(\sqrt{\theta n d / \delta})$. We have the following claim:

$$\Pr[E] \geq 1 - 0.02\delta. \tag{5}$$

Note that $E[X] = \theta n$ and hence, $\Pr[X \leq 100\theta n / \delta] \geq 1 - 0.01\delta$ by Markov's inequality. Hence, we only need to prove $\Pr[E \mid X \leq 100\theta n / \delta]$ for Claim 5. Also note that $\sum_{p \in P} \xi_p$ has the same distribution as $N(0, X \cdot I_d)$. Then by (Vershynin, 2018, Theorem 3.1.1),

$$\Pr\left[ \|\sum_{p \in P} \xi_p\|_2 \leq 10\sqrt{\theta n d / \delta} + O(\sqrt{\theta n / \delta}) \mid X \leq 100\theta n \right] \geq 1 - 0.01\delta.$$

Thus, we prove Claim 5.

In the remaining proof, we condition on event $E$. Fix an arbitrary center $c \in \mathbb{R}^d$ and let $l = \|c - c^\star\|_2$. By the optimality of $c^\star$, it is well known that
$$\mathrm{cost}(P, c) = \mathrm{cost}(P, c^\star) + n \cdot \|c - c^\star\|_2^2 = \mathsf{OPT} + n l^2.$$
Note that $|\sum_{p \in P} \langle \xi_p, p - c \rangle| \leq |\sum_{p \in P} \langle \xi_p, p - c^\star \rangle| + |\sum_{p \in P} \langle \xi_p, c - c^\star \rangle|$. By Claim 3.6 and Chebyshev's inequality, we have

$$\Pr_{\widehat{P}}[|\sum_{p \in P} \langle \xi_p, p - c^\star \rangle| \geq 10\sqrt{\theta \cdot \mathsf{OPT} / \delta}] \leq \frac{\theta \cdot \mathrm{cost}(P, c^\star)}{(10\sqrt{\theta \cdot \mathsf{OPT} / \delta})^2} = 0.01\delta. \tag{6}$$

We also have

$$\begin{aligned}
|\sum_{p \in P} \langle \xi_p, c - c^\star \rangle| &\leq \|\sum_{p \in P} \xi_p\|_2 \|c - c^\star\| \quad \text{(Cauchy-schwarz)} \\
&\leq O(l \cdot \sqrt{\theta n d \delta})
\end{aligned} \tag{7}$$

Combining with Inequalities 6 and 7, we conclude that

$$\frac{|\sum_{p \in P} \langle \xi_p, p - c \rangle|}{\mathrm{cost}(P, c)} \leq O\left( \frac{l \cdot \sqrt{\theta n d / \delta}}{\mathsf{OPT} + n l^2} \right) \leq O\left( \sqrt{\frac{\theta d}{\delta \mathsf{OPT}}} \right),$$

happens with probability at least 0.95, which completes the proof of Lemma D.2. $\square$

Overall, Inequality 4 holds with probability at least 0.9, which implies that $\widehat{P}$ is an $O(\frac{\theta n d}{\mathsf{OPT}} + \sqrt{\frac{\theta d}{\mathsf{OPT}}})$-coreset of $P$ for 1-MEANS.

**2. Proof for** $\mathsf{Err}(\cdot)$ **for** $k$**-Means.** Note that we only need to prove the following lemma.

**Lemma D.3** *With probability at least 0.8,* $\widehat{P}$ *is an* $O(\frac{\theta nd}{\mathsf{OPT}} + \theta\sqrt{\frac{\theta nd}{\mathsf{OPT}}})$*-coreset of* $P$ *for* $k$*-Means.*

**Proof:** By Claim 3.6, we have that with probability at least 0.8, $\sum_{p \in P} \|\xi_p\|_2^2 = O(\theta nd)$, which we assume happens in the following. Fix a $k$-center set $C \in \mathcal{C}$. It suffices to prove that

$$|\mathrm{cost}(P,C) - \mathrm{cost}(\widehat{P},C)| \leq O(\frac{\theta nd}{\mathsf{OPT}} + \theta\sqrt{\frac{\theta nd}{\mathsf{OPT}}}) \cdot \mathrm{cost}(P,C). \qquad (8)$$

By the triangle inequality, we know that for each $p \in P$, $|d(p,C) - d(\widehat{p},C)| \leq \|\xi_p\|_2$, which implies that $|d^2(p,C) - d^2(\widehat{p},C)| \leq \|\xi_p\|_2^2 + 2\|\xi_p\|_2 \cdot d(p,C)$. Thus, we have

$$
\begin{aligned}
&\frac{|\mathrm{cost}(P,C) - \mathrm{cost}(\widehat{P},C)|}{\mathrm{cost}(P,C)} \\
\leq\ & \frac{\sum_{p \in P} \|\xi_p\|_2^2 + 2\|\xi_p\|_2 \cdot d(p,C)}{\mathrm{cost}(P,C)} \\
\leq\ & O(\frac{\theta nd}{\mathsf{OPT}}) + \frac{2\sum_{p \in P} \|\xi_p\|_2 \cdot d(p,C)}{\mathrm{cost}(P,C)} && \text{(by assumption)} \\
\leq\ & O(\frac{\theta nd}{\mathsf{OPT}}) + \frac{2\sqrt{(\sum_{p \in P} \|\xi_p\|_2^2) \cdot (\sum_{p \in P} d^2(p,C))}}{\mathrm{cost}(P,C)} && \text{(Cauchy-Schwarz)} \\
\leq\ & O(\frac{\theta nd}{\mathsf{OPT}}) + \sqrt{\frac{O(\theta nd)}{\mathrm{cost}(P,C)}} && \text{(by assumption)} \\
\leq\ & O\left(\frac{\theta nd}{\mathsf{OPT}} + \sqrt{\frac{\theta nd}{\mathsf{OPT}}}\right), && \text{(Defn. of OPT)}
\end{aligned}
$$

which completes the proof of Inequality 8. $\qquad\square$

**3. Proof for** $\mathsf{Err}_{\mathrm{AR}}(\cdot)$ **for** **1-Means.** Now we prove the approximation-ratio coreset guarantee. Let $\mu(\widehat{P})$ denote the mean point of $\widehat{P}$. Let $\widehat{\mathsf{OPT}}$ denote the optimal 1-Means value of $\widehat{P}$. We have the following well-known properties of 1-Means:

1. $\mu(\widehat{P}) = c^\star + \frac{1}{n} \cdot \sum_{p \in P} \xi_p$;

2. $\widehat{\mathsf{OPT}} = \mathrm{cost}(\widehat{P}, \mu(\widehat{P}))$;

3. for every $c \in \mathbb{R}^d$, $\mathrm{cost}(P,c) = \mathsf{OPT} + n \cdot \|c - c^\star\|_2^2$;

4. for every $\alpha$-approximate solution $c \in \mathbb{R}^d$ of $\widehat{P}$, we have $\|\mu(\widehat{P}) - c\|_2 = \sqrt{\frac{(\alpha-1)\widehat{\mathsf{OPT}}}{n}}$.

By the above properties, we also have

$$
\begin{aligned}
\widehat{\mathsf{OPT}} = \quad & \mathrm{cost}(\widehat{P}, \mu(\widehat{P})) \\
= \quad & \sum_{\widehat{p} \in \widehat{P}} \|\widehat{p} - \mu(\widehat{P})\|_2^2 \\
= \quad & \sum_{p \in P} \|p + \xi_p - c^\star - \frac{1}{n} \cdot \sum_{q \in P} \xi_q\|_2^2 \\
= \quad & \sum_{p \in P} \left( \|p - c^\star\|_2^2 + 2\langle \xi_p, p - c^\star \rangle + \|\xi_p - \frac{1}{n} \cdot \sum_{q \in P} \xi_q\|_2^2 \right) - \frac{2}{n} \cdot \langle \sum_{p \in P} \xi_p, \sum_{p \in P} p - c^\star \rangle \\
= \quad & \mathsf{OPT} + \sum_{p \in P} \|\xi_p\|_2^2 - \frac{1}{n} \| \sum_{p \in P} \xi_p \|_2^2 + 2 \sum_{p \in P} \langle \xi_p, p - c^\star \rangle - \frac{2}{n} \cdot \langle \sum_{p \in P} \xi_p, \sum_{p \in P} p - c^\star \rangle \\
\leq \quad & \mathsf{OPT} + \sum_{p \in P} \|\xi_p\|_2^2 + 2 \sum_{p \in P} \langle \xi_p, p - c^\star \rangle + \frac{2}{n} \cdot \| \sum_{p \in P} \xi_p \|_2 \cdot \| \sum_{p \in P} p - c^\star \|_2.
\end{aligned}
$$

We first claim that with probability at least 0.99,

$$
\widehat{\mathsf{OPT}} \leq \mathsf{OPT} + O(\sqrt{\theta d \cdot \mathsf{OPT}} + \theta n d) \leq O(\mathsf{OPT} + \theta n d). \tag{9}
$$

By Claim D.1, we know that $\sum_{p \in P} \|\xi_p\|_2^2$ concentrates on $\theta n d$. By Lemma D.2, we know that

$$
|\sum_{p \in P} \langle \xi_p, p - c^\star \rangle| \leq \mathrm{cost}(P, c^\star) \cdot O(\sqrt{\frac{\theta d}{\mathsf{OPT}}}) = O(\sqrt{\theta d \cdot \mathsf{OPT}}).
$$

We also note that $\mathbb{E}_{\widehat{P}}\left[ \sum_{p \in P} \xi_p \right] = 0$ and

$$
\begin{aligned}
\mathrm{Var}_{\widehat{P}}\left[ \| \sum_{p \in P} \xi_p \|_2 \right] \leq \quad & \mathbb{E}_{\widehat{P}}\left[ \| \sum_{p \in P} \xi_p \|_2^2 \right] \\
= \quad & \mathbb{E}_{\widehat{P}}\left[ \sum_{p \in P} \|\xi_p\|_2^2 \right] \\
= \quad & \theta n d,
\end{aligned} \tag{10}
$$

which implies that $\| \sum_{p \in P} \xi_p \|_2$ is $O(\sqrt{\theta n d})$ with probability at least 0.999. Thus, we have that with probability at least 0.995,

$$
\frac{2}{n} \cdot \| \sum_{p \in P} \xi_p \|_2 \cdot \| \sum_{p \in P} p - c^\star \|_2 \leq \frac{2}{n} \cdot O(\sqrt{\theta n d}) \cdot \sqrt{n \cdot \mathsf{OPT}} \leq O(\theta d \cdot \mathsf{OPT}).
$$

Overall, we prove Inequality 9.

By Definition 2.2, it suffices to prove that for any $(1 + \alpha)$-approximate solution $c \in \mathbb{R}^d$ of $\widehat{P}$, the following inequality holds:

$$
\begin{aligned}
\frac{r_P(C) - r_{\widehat{P}}(C)}{r_{\widehat{P}}(C)} = \quad & \frac{n \cdot \|c^\star - c\|_2^2 \cdot \widehat{\mathsf{OPT}} - n \cdot \|\mu(\widehat{P}) - c\|_2^2 \cdot \mathsf{OPT}}{\mathsf{OPT} \cdot \mathrm{cost}(\widehat{P}, c)} \\
\leq \quad & O\left( \frac{\theta d}{\mathsf{OPT}} + \frac{(1 - \frac{1}{\alpha})\theta n d}{\mathsf{OPT}} + (1 - \frac{1}{\alpha}) \right).
\end{aligned} \tag{11}
$$

Assuming Inequality 9 holds and letting $\beta = r_{\widehat{P}}(c) \in [1, \alpha]$, we have

$$
\frac{n \cdot \|c^\star - c\|_2^2 \cdot \widehat{\mathsf{OPT}} - n \cdot \|\mu(\widehat{P}) - c\|_2^2 \cdot \mathsf{OPT}}{\mathsf{OPT} \cdot \mathrm{cost}(\widehat{P}, c)}
$$

$$
\leq \quad \frac{n \cdot \|c^\star - c\|_2^2 \cdot \widehat{\mathsf{OPT}}}{\mathsf{OPT} \cdot \beta \cdot \widehat{\mathsf{OPT}}}
$$

$$
\leq \quad \frac{2n \cdot (\|c^\star - \mu(\widehat{P})\|_2^2 + \|\mu(\widehat{P}) - c\|_2^2)}{\beta \cdot \mathsf{OPT}}
$$

$$
\leq \quad \frac{2\|\sum_{p \in P} \xi_p\|_2^2}{n \cdot \mathsf{OPT}} + \frac{2(\beta - 1) \cdot \widehat{\mathsf{OPT}}}{\beta \cdot \mathsf{OPT}} \qquad \left( \mu(\widehat{P}) = c^\star + \frac{1}{n} \cdot \sum_{p \in P} \xi_p \right)
$$

$$
\leq \quad O(\frac{\theta d}{\mathsf{OPT}}) + 2(\beta - 1) \cdot \frac{O(\mathsf{OPT} + \theta n d)}{\beta \cdot \mathsf{OPT}}, \qquad \text{(Ineq. 9)}
$$

$$
\leq \quad O(\frac{\theta d}{\mathsf{OPT}} + \frac{(1 - \frac{1}{\beta})\theta n d}{\mathsf{OPT}} + (1 - \frac{1}{\beta}))
$$

$$
\leq \quad O(\frac{\theta d}{\mathsf{OPT}} + \frac{(1 - \frac{1}{\alpha})\theta n d}{\mathsf{OPT}} + (1 - \frac{1}{\alpha})), \qquad (\beta \leq \alpha)
$$

which verifies Inequality 11. Hence, we complete the proof of Lemma 3.5 for $k = 1$.

**4. Proof for $\mathrm{Err}_{\mathrm{AR}}(\cdot)$ for $k$-MEANS.** Finally, we show how to extend to general $k \geq 1$. Let $C^\star \in \mathcal{C}$ denote an optimal solution for $k$-MEANS of $P$. First, similar to the 1-MEANS setting, we have the following guarantee on the optimal clustering cost of $\widehat{P}$.

**Lemma D.4 (Bounding $\widehat{\mathsf{OPT}}$)** *With probability at least 0.9, we have for all $i \in [k]$*

$$
\mathrm{cost}(\widehat{P}_i, \hat{c}_i) \leq \mathsf{OPT}_i + O\left(\theta n_i d k + \sqrt{\theta d k \cdot \mathsf{OPT}_i}\right) \leq 1.5\mathsf{OPT}_i + O(\theta n_i d k).
$$

*Besides, we also have with high probability*

$$
\widehat{\mathsf{OPT}} \leq \sum_i \mathrm{cost}(\widehat{P}_i, \hat{c}_i) \leq \mathsf{OPT} + O(\theta n d + \sqrt{\theta d \cdot \mathsf{OPT}})
$$

**Proof:** Recall that we use $P_i$ to denote the data clustered to $c_i$ in $P$, where $c_i$ is the $i$-th center in the optimal cluster of $P$. We use $\widehat{P}_i$ to denote the points in $P_i$ with presence of the noise, and $\hat{c}_i$ to denote the mean of $P_i$.

Similar to the 1-MEANS setting (Claim 3.6), we have the following decomposition of the error

$$
\mathrm{cost}(\widehat{P}_i, \hat{c}_i) - \mathrm{cost}(P_i, c_i) = \sum_{p \in P_i} \|\xi_p\|_2^2 + 2 \sum_{p \in P_i} \langle \xi_p, p - c_i \rangle.
$$

Besides, we have the following variance of the error terms

- $\mathbb{E}_{\widehat{P}_i}\left[ \sum_{p \in P_i} \|\xi_p\|_2^2 \right] = \theta n_i d$ and $\mathrm{Var}_{\widehat{P}_i}\left[ \sum_{p \in P_i} \|\xi_p\|_2^2 \right] = O(\theta n_i d^2)$;

- $\mathbb{E}_{\widehat{P}_i}\left[ \sum_{p \in P_i} \langle \xi_p, p - c_i \rangle \right] = 0$ and $\mathrm{Var}_{\widehat{P}_i}\left[ \sum_{p \in P_i} \langle \xi_p, p - c_i \rangle \right] = \theta \cdot \mathsf{OPT}_i$.

From Lemma D.2 by choosing $\delta = 1/k$, we know that with probability at least $1 - 0.05/k$, we have

$$
\sup_{c \in \mathbb{R}^d} \frac{|\sum_{p \in P_i} \langle \xi_p, p - c_i \rangle|}{\mathrm{cost}(P_i, c_i)} = O\left( \sqrt{\frac{\theta d k}{\mathsf{OPT}_i}} \right).
$$

Besides from Chybeshev's inequality, we also have $\sum_{p \in P_i} \|\xi_p\|^2 \leq O(\theta n_i d k)$ happens with probability at least $1 - 0.05/k$. Then we conclude the proof by applying union bound on $i \in [k]$. $\square$

Then given the well-seperated data assumption, we have the following.

**Lemma D.5 (Structural property of $\mathcal{C}_\alpha(\widehat{P})$)** *Given $1 \leq \alpha \leq 1 + \frac{1}{k}$. For every $C \in \mathcal{C}_\alpha(\widehat{P})$ and every $i \in [k]$, there must exist a center $c \in C$ such that $c \in B(c_i^\star, O(\sqrt{k}a_i + \sqrt{kd}))$.*

**Proof:** Denote $P_i' = \widehat{P}_i \cap B(c_i^\star + O(\sqrt{kd}))$ as the points in $\widehat{P}_i$ within the ball $B(c_i^\star, a_i + O(\sqrt{kd}))$. We first show that, with high probability, for all $1 \leq i \leq k$, $|\widehat{P}_i| \geq 0.5n_i$. Note that for fixed $p$, we have $\mathbb{E}\|\xi_p\|^2 = \theta d$ and $\mathrm{Var}\|\xi_p\|^2 = \theta d^2$. Thus by Chebyshev's inequality,

$$
\begin{aligned}
\Pr\left[\|\xi_p\|^2 > 4d\right] &\leq \quad \Pr\left[\|\xi_p\|^2 - \theta d > 3d\right] \\
&\leq \quad \frac{\theta d^2}{9d^2} \\
&\leq \quad \theta/9.
\end{aligned}
$$

Thus for a single point $p \in P_i$, with probability at most $1/9$, $\hat{p} \notin B(c_i^\star + O(\sqrt{kd}))$. Then with Chernoff bound, as long as $n_i \geq O(\log k)$, we have

$$
\Pr\left[|P_i'| \leq n_i - 0.5n_i\right] \leq \exp\left(-\delta^2\mu/(2+\delta)\right),
$$

where $\delta = 0.5n_i/\mu - 1$ and $\mu \leq \theta * n_i/9$. By computation, we have $\delta \geq 4.5/\theta - 1$, and $\delta\mu = 0.5n_i - \mu \geq 0.35n_i$. Thus

$$
\exp\left(-\delta^2\mu/2\right) \leq \exp\left(-\delta\mu/2\right) \leq \exp(0.35n_i) \leq 0.01,
$$

as long as $n_i = O(\log k)$. Thus we know that with high probability, for all $1 \leq i \leq k$, $|\widehat{P}_i| \geq 0.5n_i$.

Next we prove the lemma by contradiction. For simplicity, we denote $\gamma \geq \max_{i,j} \frac{\mathsf{OPT}_i}{\mathsf{OPT}_j} + \max_{i,j} \frac{n_i}{n_j}$, which is bounded by some constant. Assume that for all $1 \leq j \leq k$, $|\widehat{P}_j| \geq 0.5n_j$, but $C \cap B(c_i^\star, 4\sqrt{k}\sqrt{\gamma}a_i + O(\sqrt{kd})) = \phi$. Let $P' \subseteq \widehat{P}_i$ denote the collection of points within $B(c_i^\star, 4\sqrt{k}\sqrt{\gamma}a_i + O(\sqrt{kd}))$. We have

$$
\begin{aligned}
\mathrm{cost}(P', C) &\geq \quad 0.5n_i \cdot (3\sqrt{k}\sqrt{\gamma}a_i + O(\sqrt{kd}))^2 \\
&\geq \quad 4.5n_i(k\gamma a_i^2 + O(\sqrt{kd})) \\
&\geq \quad 4.5k\gamma\mathsf{OPT}_i + O(n_i kd) \\
&\geq \quad 4.5\mathsf{OPT} + O(n_i kd).
\end{aligned}
$$

Note that from Lemma D.4, we have with high probability, for every $j \in [k]$, we have

$$
\mathrm{cost}(\widehat{P}_j, \hat{c}_j) \leq 1.5\mathsf{OPT}_j + O(\theta n_j dk),
$$

and

$$
\widehat{\mathsf{OPT}} \leq \sum_i \mathrm{cost}(\widehat{P}_i, \hat{c}_i) \leq \mathsf{OPT} + O(\theta nd + \sqrt{\theta d \cdot \mathsf{OPT}}).
$$

Thus with high probability,

$$
\frac{\mathrm{cost}(\hat{P}, C)}{\widehat{\mathsf{OPT}}} \geq \frac{\mathrm{cost}(P', C)}{\widehat{\mathsf{OPT}}} \geq \frac{4.5\mathsf{OPT} + O(n_i kd)}{\mathsf{OPT} + O(nkd)} \geq 1 + \alpha,
$$

which concludes the proof. $\qquad\qquad\square$

**Lemma D.6 (Controlling assignment difference between $P$ and $\widehat{P}$)** *For any $C$ satisfies for all $i$, there exists $c_i \in C \cap B(c_i^\star, O(\sqrt{k}a_i + \sqrt{kd}))$, we have with high probability (say $0.99$)*

$$
\sum_i \mathrm{cost}(\widehat{P}_i, c_i) - \mathrm{cost}(\widehat{P}, C) = \exp\left(-O(\sqrt{kd})\right) \cdot \theta n.
$$

**Proof:** We first compute the expectation for a fixed $C$

$$\mathbb{E}_{\widehat{P}}\left[\sum_i (\text{cost}(\widehat{P}_i, c_i) - \text{cost}(\widehat{P}, C))\right].$$

For simplicity, we use $B_i$ to denote the ball $B(c_i^\star, O(\sqrt{k}a_i + \sqrt{kd}))$. Note that if for $p \in \widehat{P}_i$ and $p \in B_i$,

$$\|p - c_i\| < \|p - c_j\|, \forall j \neq i.$$

Thus, we only need to consider the points such that $p \notin \cup_i B_i$.

Note that since $\xi_p$ satisfies Bernstein condition if $\xi_p \neq 0$, we have

$$\Pr\left[|\xi_p| \geq t | \xi_p \neq 0\right] \leq 2\exp\left(-\frac{t^2}{2(1 + bt)}\right), \forall t > 0.$$

Thus we have for large enough $t$ (larger than $b$),

$$\Pr\left[|\xi_p| \geq t | \xi_p \neq 0\right] \leq 2\exp\left(-O(t)\right).$$

Then we can decompose the cost as

$$\mathbb{E}\left[\sum_i \text{cost}(\widehat{P}_i, c_i) - \text{cost}(\widehat{P}, C)\right]$$

$$\leq \sum_i n_i \cdot \theta \cdot \int_{O(\sqrt{k}a_i + \sqrt{kd})} \left(O(\sqrt{k}a_i + \sqrt{kd}) + t\right)^2 \rho(\xi_p = t)\mathrm{d}t$$

$$\leq \sum_i n_i \cdot \theta \cdot \int_{O(\sqrt{k}a_i + \sqrt{kd})} 4t^2 \rho(\xi_p = t)\mathrm{d}t.$$

Note that

$$\int_{(\Gamma - 1)a_i + O(\sqrt{kd})} t^2 \rho(\xi_p = t)\mathrm{d}t = \mathbb{E}[X] = \int_0^\infty \Pr[X \geq s]\mathrm{d}s,$$

where $X = 0$ when $t \leq (\Gamma - 1)a_i + O(\sqrt{kd})$ and $X = t^2$ otherwise. Note that when $s \leq ((\Gamma - 1)a_i + O(\sqrt{kd}))^2$, $\Pr[X \geq s] \leq \exp\left(-O((\Gamma - 1)a_i + O(\sqrt{kd}))\right)$ and when $s > ((\Gamma - 1)a_i + O(\sqrt{kd}))^2$, we have

$$\Pr[X > s] \leq \exp\left(-O(\sqrt{s})\right).$$

Also note that

$$\int \exp\left(-o(\sqrt{s})\right)\mathrm{d}s = -O(1) \cdot \exp\left(-O(\sqrt{x})\right) \cdot (\sqrt{x} + O(1)) + \text{Constant}.$$

Thus we have

$$\mathbb{E}[X] \leq O(1) \cdot \exp\left(-O((\Gamma - 1)a_i + O(\sqrt{kd}))\right) \cdot (O((\Gamma - 1)a_i + O(\sqrt{kd})) + O(1)) = o(\frac{1}{kd}).$$

Then we complete the proof by applying Markov's inequality. $\qquad\square$

Now we can finish our proof for the $\text{Err}_{\text{AR}}(\cdot)$ guarantee in Lemma 3.5.

**Proof:** [of $\text{Err}_{\text{AR}}(\cdot)$ guarantee in Lemma 3.5] It suffices to show that for any solution $C$, we have with high probability

$$\frac{r_P(C) - r_{\widehat{P}}(C)}{r_{\widehat{P}}(C)} = \frac{\frac{\text{cost}(P,C)}{\text{OPT}} - \frac{\text{cost}(\widehat{P},C)}{\widehat{\text{OPT}}}}{\frac{\text{cost}(\widehat{P},C)}{\widehat{\text{OPT}}}} \leq O(\frac{\theta d}{\text{OPT}} + \frac{(1 - \frac{1}{\alpha} + \exp\left(-O(\sqrt{kd})\right))\theta nd}{\text{OPT}} + (1 - \frac{1}{\alpha}).$$

Note that with high probability, we have

$$\widehat{\text{OPT}} \leq \sum_i \text{cost}(\widehat{P}_i, \hat{c}_i) \leq \text{OPT} + O(\theta nd + \sqrt{\theta d \cdot \text{OPT}}). \tag{12}$$

Thus we can assume that Eq. 12 holds. Thus, we have

$$\widehat{\mathsf{OPT}} \le O(\mathsf{OPT} + \theta nd).$$

Besides, since Lemma D.5 holds with high probability, we can also assume that for every $i \in [k]$, there exists $c \in C$ such that $c \in B(c_i^\star, O(\sqrt{k}a_i + \sqrt{kd}))$, and we denote $c_i \in C$ such that $c_i \in B(c_i^\star, \Gamma a_i + O(\sqrt{kd}))$. Note that from the assumption on data, we know that $B(c_i^\star, O(\sqrt{k}a_i + \sqrt{kd})) \cap B(c_j^\star, O(\sqrt{k}a_i + \sqrt{kd})) = \phi$, for all $i \neq j$, and thus $c_i$ is well-defined. Besides, we also know that for all $p \in P_i$, $\|p - c_i\| \le \|p - c_j\|$ for all $j \neq i$.

Then denote $\beta = r_{\widehat{P}}(c) \in [1, \alpha]$, we have

$$\frac{r_P(C) - r_{\widehat{P}}(C)}{r_{\widehat{P}}(C)}$$

$$\le \frac{\mathrm{cost}(P,C) - \mathsf{OPT}}{\beta \mathsf{OPT}}$$

$$= \frac{\sum_{i=1}^k (\mathrm{cost}(P_i, c_i) - \mathrm{cost}(P_i, c_i^\star))}{\beta \mathsf{OPT}}$$

$$= \frac{\sum_{i \in [k]} n_i \cdot \|c_i^\star - c_i\|^2}{\beta \sum_{i \in [k]} \mathrm{cost}(P_i, c_i^\star)}$$

$$\le \frac{\sum_{i=1}^k \left( 2n_i \cdot (\|c_i^\star - \mu(\widehat{P}_i)\|_2^2 + \|\mu(\widehat{P}_i) - c_i\|_2^2) \right)}{\beta \cdot \sum_{i \in [k]} \mathrm{cost}(P_i, c_i^\star)}$$

$$\le \frac{\sum_i \frac{1}{n_i} 2\|\sum_{p \in P_i} \xi_p\|_2^2}{\beta \cdot \mathsf{OPT}}$$

$$+ \frac{2(\sum_i \mathrm{cost}(\widehat{P}_i, c_i) - \mathrm{cost}(\widehat{P}, C) + \mathrm{cost}(\widehat{P}, C) - \sum_i \mathrm{cost}(\widehat{P}_i, \mu(\widehat{P}_i))}{\beta \cdot \sum_{i \in [k]} \mathrm{cost}(P_i, c_i^\star)}$$

$$\le O(\frac{\theta kd}{\mathsf{OPT}}) + \frac{2 \exp\left(-O(\sqrt{kd})\right) \theta dn}{\beta \cdot \sum_{i \in [k]} \mathrm{cost}(P_i, c_i^\star)} + \frac{2(\beta - 1)\widehat{\mathsf{OPT}}}{\beta \cdot \sum_{i \in [k]} \mathrm{cost}(P_i, c_i^\star)}$$

$$\le O(\frac{\theta kd}{\mathsf{OPT}}) + 2(\beta - 1) \cdot \frac{O(\mathsf{OPT} + \theta nd)}{\beta \cdot \mathsf{OPT}} + + \frac{2 \exp\left(-O(\sqrt{kd})\right) \theta dn}{\beta \mathsf{OPT}}, \qquad \text{(Ineq. 9)}$$

$$\le O(\frac{\theta kd}{\mathsf{OPT}} + \frac{(1 - \frac{1}{\beta} + \exp\left(-O(\sqrt{kd})\right))\theta nd}{\mathsf{OPT}} + (1 - \frac{1}{\beta}))$$

$$\le O(\frac{\theta kd}{\mathsf{OPT}} + \frac{(1 - \frac{1}{\alpha} + \exp\left(-O(\sqrt{kd})\right))\theta nkd}{\mathsf{OPT}} + (1 - \frac{1}{\alpha})). \qquad (\beta \le \alpha)$$

Thus we conclude the proof of Theorem 3.1.

$\square$

## D.2 PROOF OF THEOREM 3.3: ERROR LOWER BOUNDS FOR 1-MEANS CORESET

We only consider $c^\star$ for the coreset error. By Claim 3.6 and Lemma D.2, we observe that $\sum_{p \in P} \|\xi_p\|_2^2$ is highly concentrated at its expectation $\theta nd$, which implies that with probability at least 0.8,

$$\mathrm{cost}(\widehat{P}, c^\star) - \mathrm{cost}(P, c^\star) = \Omega(\theta nd).$$

Consequently, $\widehat{P}$ is an $\Omega(\frac{\theta nd}{\mathsf{OPT}})$-coreset of $P$.

For the error of approximation-ratio coreset, we first recall that

$$\widehat{\mathsf{OPT}} = \mathsf{OPT} + (1 - \frac{1}{n})\sum_{p \in P} \|\xi_p\|_2^2 + 2\sum_{p \in P} \langle \xi_p, p - c^\star \rangle - \frac{2}{n} \cdot \langle \sum_{p \in P} \xi_p, \sum_{p \in P} p - c^\star \rangle.$$

With probability 0.5, we have $2\sum_{p\in P}\langle\xi_p, p - c^\star\rangle - \frac{2}{n}\cdot\langle\sum_{p\in P}\xi_p, \sum_{p\in P}p - c^\star\rangle \geq 0$. Then again by Claim 3.6, we conclude that $\sum_{p\in P}\|\xi_p\|_2^2 = \Theta(\theta nd)$ and $\widehat{\mathsf{OPT}} = \mathsf{OPT} + \Omega(\theta nd)$ with probability at least 0.3. Next, we consider center $c = \mu(\widehat{P}) + \sqrt{\frac{(\alpha-1)\cdot\widehat{\mathsf{OPT}}}{n}}\cdot\frac{\mu(\widehat{P})-c^\star}{\|\mu(\widehat{P})-c^\star\|_2}$. By the proof of Lemma 3.5, we know that

$$\mathrm{cost}(\widehat{P}, c) = \alpha\cdot\widehat{\mathsf{OPT}},$$

i.e., $c$ is an $\alpha$-approximate solution of $S$. We also have

$$
\begin{aligned}
\mathrm{cost}(P, c) =\ & \mathsf{OPT} + n\cdot\|c - c^\star\|_2^2 \\
=\ & \mathsf{OPT} + n\cdot\left(\|\frac{1}{n}\cdot\sum_{p\in P}\xi_p\|_2 + \sqrt{\frac{(\alpha-1)\cdot\widehat{\mathsf{OPT}}}{n}}\right)^2 \\
\geq\ & \mathsf{OPT} + \frac{1}{n}\cdot\|\sum_{p\in P}\xi_p\|_2^2 + (\alpha-1)\cdot\widehat{\mathsf{OPT}},
\end{aligned}
$$

and

$$\mathrm{cost}(P, \mu(\widehat{P})) = \mathsf{OPT} + \frac{1}{n}\cdot\|\sum_{p\in P}\xi_p\|_2^2 = \mathsf{OPT} + \Theta(\theta d).$$

Combining with the above inequalities, we conclude that

$$
\begin{aligned}
r_P(c) \geq\ & \frac{\mathsf{OPT} + \frac{1}{n}\cdot\|\sum_{p\in P}\xi_p\|_2^2 + (\alpha-1)\cdot\widehat{\mathsf{OPT}}}{\mathsf{OPT}} \\
\geq\ & \alpha + \frac{\Omega((\alpha-1)\theta nd) - \frac{\alpha}{n}\cdot\|\sum_{p\in P}\xi_p\|_2^2}{\mathsf{OPT}} & (\widehat{\mathsf{OPT}} = \mathsf{OPT} + \Theta(\theta nd)) \\
\geq\ & \alpha + \frac{\Omega((\alpha-1)\theta nd) - O(\theta d)}{\mathsf{OPT}} & (\|\sum_{p\in P}\xi_p\|_2^2 = O(\theta nd)) \\
\geq\ & \alpha + \frac{\Omega((\alpha-1)\theta nd)}{\mathsf{OPT}} & (\alpha-1 \gg \frac{1}{n}) \\
\geq\ & \alpha + \Omega(\frac{(\alpha-1)\theta nd}{\mathsf{OPT}}) \\
\geq\ & (1 + \Omega(\frac{(\alpha-1)\theta nd}{\mathsf{OPT}}))\cdot r_{\widehat{P}}(c). & (\mathrm{cost}(\widehat{P}, c) = \alpha\cdot\widehat{\mathsf{OPT}})
\end{aligned}
$$

Hence, we conclude that $\widehat{P}$ is an $(\alpha, \Omega(\frac{(1-\frac{1}{\alpha})\theta nd}{\mathsf{OPT}}))$-approximation-ratio coreset of $P$ for 1-MEANS, which completes the proof.

### D.3 TECHNICAL CHALLENGE FOR ROBUST $k$-MEANS CORESET

Now we show it is difficult to construct an $o(\frac{\theta d}{\mathsf{OPT}})$-coreset $S$ for $k$-MEANS of $P$ from $\widehat{P}$ under noise model I. Let us first consider 1-MEANS. As demonstrated in the proof of Theorem 3.1, it becomes apparent that the optimal center $c^\star$ undergoes a displacement $\frac{1}{n}\sum_{p\in P}\xi_p$ with norm approximately $\Theta(\sqrt{\frac{\theta d}{n}})$ in a uniformly random direction when $\theta \gg \frac{1}{n}$. Consequently, any coreset $S$ constructed by $\widehat{P}$ is likely to satisfy

$$\|\mu(S) - c^\star\|_2 = \|\frac{1}{w(S)}\cdot\sum_{p\in S}w(p)\cdot p - \frac{1}{n}\sum_{p\in P}p\|_2 \geq \Omega(\sqrt{\frac{\theta d}{n}}). \tag{13}$$

Next, we have the following lemma.

**Lemma D.7 (Hardness of robust 1-MEANS coreset)** *Let $S \subset \mathbb{R}^d$ be a weighted dataset satisfying Inequality 13 and $\mu(S) = \frac{1}{w(S)}\sum_{p\in S}w(p)\cdot p$ be the optimal center of $S$ for 1-MEANS. There we have $\mathrm{cost}(P, \mu(S)) \geq \mathsf{OPT} + \Omega(\theta d)$.*

The above lemma shows that $\mu(S)$ is a $1 + \Omega(\frac{\theta d}{\mathsf{OPT}})$-approximate center of $P$, which impedes our ability to construct a noise-robust coreset for arbitrary small $\varepsilon > 0$.

**Proof:** [of Lemma D.7] By the proof of Lemma 3.5, we know that for every $c \in \mathbb{R}^d$,

$$\mathrm{cost}(P, c) = \mathsf{OPT} + n \cdot \|c^\star - c\|_2^2.$$

Letting $c = \mu(S)$, we have

$$\mathrm{cost}(P, \mu(S)) = \mathsf{OPT} + n \cdot \|c^\star - \mu(S)\| \geq \mathsf{OPT} + \Omega(\theta d),$$

by the assumption of the lemma. This completes the proof. $\qquad\square$

## E   RESULTS FOR $(k, z)$-CLUSTERING CORESETS

In this section, we provide theoretical results for the $(k, z)$-CLUSTERING problem (Theorems E.1 and E.2). These results do not make assumptions on data and hence, the bounds for $\mathsf{Err}_{\mathrm{AR}}(\cdot)$ are weaker than that of Theorem 3.1. Let $C^\star \in \mathbb{R}^d$ denote an optimal solution of $P$. Let $\mathsf{OPT}$ denote the optimal $(k, z)$-CLUSTERING value of the underlying dataset $P$. We first have the following theorem under noise model I.

**Theorem E.1 (Performance analysis for $(k, z)$-CLUSTERING coreset under noise model I)** *Let $\varepsilon, \theta \in [0, 1]$ and $z \geq 1$ be constant. Let $P \subset \mathbb{R}^d$ be an underlying dataset of size $n \geq 1$ and $\widehat{P} \subset \mathbb{R}^d$ be an observed dataset drawn from $P$ under the noise model I with underlying parameter $\theta$. Let $S$ be an $\varepsilon$-coreset of $\widehat{P}$ for $(k, z)$-CLUSTERING. With probability at least 0.8, $S$ is an $O(\varepsilon + \frac{\theta n d^{z/2}}{\mathsf{OPT}} + \sqrt[z]{\frac{\theta n d^{z/2}}{\mathsf{OPT}}})$-coreset of $P$ for $(k, z)$-CLUSTERING.*

*Moreover, there exists $\theta \in [0, 1]$ and a dataset $P \subset \mathbb{R}^d$ of size $n$ such that with probability at least 0.8, $\widehat{P}$ is an $\Omega(\frac{\theta n d^{z/2}}{\mathsf{OPT}})$-coreset and an $(+\infty, \Omega(\frac{\theta n d^{z/2}}{\mathsf{OPT}}))$-approximation-ratio coreset of $P$ for $(k, z)$-CLUSTERING when $k = n - 1$.*

By Definition 2.2, the theorem implies that for any $\alpha \geq 1$, $\mathsf{Err}_{\mathrm{AR}}(S) = O(\varepsilon + \frac{\theta n d^{z/2}}{\mathsf{OPT}} + \sqrt[z]{\frac{\theta n d^{z/2}}{\mathsf{OPT}}})$, which is much weaker than that of Theorem 3.1. Further improving this bound is an interesting open problem.

**Proof:** We first take $k$-MEANS as an example and then show how to extend to general $(k, z)$-CLUSTERING.

**Upper bound.** The upper bound is already proved by Lemma D.3 in Theorem 3.1.

**Worst-case lower bound.** In symmetry, we only need to prove the lower error bound $\Omega(\frac{\theta n d}{\mathsf{OPT}})$ for approximation-ratio coreset. Let $\theta = 0.1$, $n = 10000$ and $k = n - 1$. We first give the construction of $P$: let $P = \{p_i = 100 n e_i : i \in [n]\} \subset \mathbb{R}^n$ where $e_i$ is the $i$-th unit basis in $\mathbb{R}^n$. An optimal solution $C^\star = \left\{p_1, \ldots, p_{n-2}, \frac{p_{n-1} + p_n}{2}\right\}$, and hence, $\mathsf{OPT} = 10000 n^2$.

Next, we consider $\widehat{P}$. With probability at least 0.8, the following events hold:

1. $\sum_{p \in P} \|\xi_p\|_2^2 = \Theta(\theta n d)$;

2. for every $p \in P$, $\|\xi_p\|_2^2 \leq 10 d \log n$.

Event 2 is due to noise model I, which ensures that $\Pr_{\widehat{P}}\left[\|\xi_p\|_2^2 \geq 10 d \log n\right] \leq \frac{1}{10 n^2}$. Then by the union bound, event 2 happens with probability at least $1 - 0.1 n$. Conditioned on these events, the optimal solution $\widehat{C} \in \mathcal{C}$ of $\widehat{P}$ must consist of $n - 2$ points $\widehat{p} \in \widehat{P}$ and the average of the remaining

two points in $\widehat{P}$. W.l.o.g., assume $\widehat{C} = \left\{ \widehat{p}_1, \ldots, \widehat{p}_{n-2}, \frac{\widehat{p}_{n-1}+\widehat{p}_n}{2} \right\}$. Then we have

$$
\begin{aligned}
\text{cost}(P, \widehat{C}) &\geq \sum_{i \in [n-2]} \|\widehat{p}_i - p_i\|_2^2 + \text{OPT} && \text{(Defn. of OPT)} \\
&\geq \text{OPT} + \sum_{p \in P} \|\xi_p\|_2^2 - 20d \log n && \text{(Event 2)} \\
&\geq \text{OPT} + \Theta(\theta n d) - 20d \log n && \text{(Event 1)} \\
&\geq (1 + \Omega(\frac{\theta n d}{\text{OPT}})) \cdot \text{OPT}, && (\theta = 0.1)
\end{aligned}
$$

which implies that

$$
\begin{aligned}
\frac{\text{cost}(P, \widehat{C})}{\text{cost}(P, C^\star)} &\geq 1 + \Omega(\frac{\theta n d}{\text{OPT}}) \\
&\geq (1 + \Omega(\frac{\theta n d}{\text{OPT}})) \cdot \frac{\text{cost}(\widehat{P}, \widehat{C})}{\text{cost}(\widehat{P}, C^\star)} && (\widehat{C} \text{ is optimal for } \widehat{P}).
\end{aligned}
$$

Thus, we complete the proof of Theorem 3.1 for $k$-MEANS.

Now we show how to extend to $(k, z)$-CLUSTERING coreset. For the upper bound, the main difference is that we have

$$
|d^z(p, C) - d^z(\widehat{p}, C)| \leq O_z \left( \|\xi_p\|_2^z + \|\xi_p\|_2 \cdot d^{z-1}(p, C) \right),
$$

where $O_z(\cdot)$ hides constant factor $2^{O(z)}$. Similar to Claim 3.6, we assume $\sum_{p \in P} \|\xi_p\|_2^z \leq O_z(\theta n d^{z/2})$ by the Bernstein condition, which happens with probability at least 0.9. Then we have

$$
\begin{aligned}
&\frac{|\text{cost}_z(P, C) - \text{cost}_z(\widehat{P}, C)|}{\text{cost}_z(P, C)} \\
\leq\ & \frac{O_z \left( \sum_{p \in P} \|\xi_p\|_2^z + \|\xi_p\|_2 \cdot d^{z-1}(p, C) \right)}{\text{cost}_z(P, C)} \\
\leq\ & O(\frac{\theta n d}{\text{OPT}}) + \frac{O_z \left( \sum_{p \in P} \|\xi_p\|_2 \cdot d^{z-1}(p, C) \right)}{\text{cost}_z(P, C)} && \text{(by assumption)} \\
\leq\ & O(\frac{\theta n d^{z/2}}{\text{OPT}}) + \frac{O_z \left( \sqrt[z]{(\sum_{p \in P} \|\xi_p\|_2^z)(\sum_{p \in P} d^z(p, C))^{z-1}} \right)}{\text{cost}_z(P, C)} && \text{(Generalized Hölder inequality)} \\
\leq\ & O(\frac{\theta n d}{\text{OPT}}) + \sqrt[z]{\frac{O(\theta n d^{z/2})}{\text{cost}_z(P, C)}} && \text{(by assumption)} \\
\leq\ & O(\frac{\theta n d}{\text{OPT}} + \sqrt[z]{\frac{\theta n d^{z/2}}{\text{OPT}}}), && \text{(Defn. of OPT)}
\end{aligned}
$$

which completes the proof of the upper bound.

For the lower bound, the argument is almost identical to that of Theorem 3.1. The only difference is that the events change to

1. $\sum_{p \in P} \|\xi_p\|_2^z = \Theta(\theta n d^{z/2})$;
2. for every $p \in P$, $\|\xi_p\|_2^z \leq (10d \log n)^{z/2}$.

Then the construction of $\widehat{C}$ is the same and we still conclude that $\widehat{P}$ is an $\Omega(\frac{\theta n d^{z/2}}{\text{OPT}})$-(weak) coreset of $P$ for $(k, z)$-CLUSTERING.

$\square$

Similarly, we give the following theorem for $(k, z)$-CLUSTERING under noise model II.

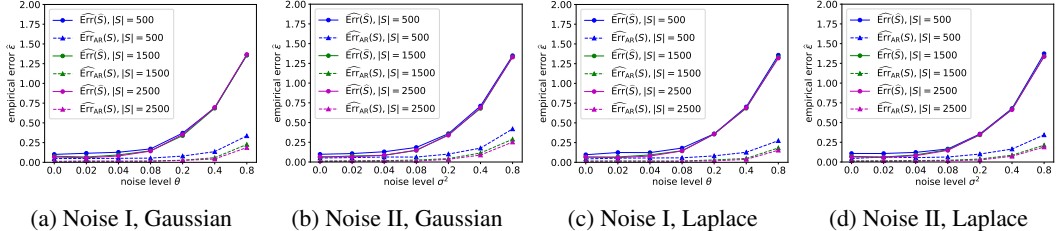

(a) Noise I, Gaussian     (b) Noise II, Gaussian     (c) Noise I, Laplace     (d) Noise II, Laplace

Figure 2: The empirical errors $\widehat{\mathrm{Err}}(S)$ and $\widehat{\mathrm{Err}}_{\mathrm{AR}}(S)$ versus the noise level $\theta$ plot for coreset $S$ with different sizes. The solid lines in each figure represent the coreset measure $\widehat{\mathrm{Err}}(S)$, and the dashed lines represent the weak coreset measure $\widehat{\mathrm{Err}}_{\mathrm{AR}}(S)$. We show the results from Adult dataset, and Figures 2a to 2d denote the noise models I, II with Gaussian and Laplace noise respectively.

**Theorem E.2 (Performance analysis for $(k, z)$-CLUSTERING coreset under noise model II)**
*Let $\varepsilon \in [0, 1], \sigma \geq 0$ and $z \geq 1$ be constant. Let $P \subset \mathbb{R}^d$ be an underlying dataset of size $n \geq 1$ and $\widehat{P} \subset \mathbb{R}^d$ be an observed dataset drawn from $P$ under the noise model II with underlying parameter $\theta$. Let $S$ be an $\varepsilon$-coreset of $\widehat{P}$ for $(k, z)$-CLUSTERING. With probability at least 0.8, $S$ is an $O(\varepsilon + \frac{\sigma^z nd^{z/2}}{\mathsf{OPT}} + \sqrt[z]{\frac{\sigma^z nd^{z/2}}{\mathsf{OPT}}})$-coreset of $P$ for $(k, z)$-CLUSTERING.*

*Moreover, there exists a dataset $P \subset \mathbb{R}^d$ of size $n$ such that with probability at least 0.8, $\widehat{P}$ is an $\Omega(\frac{\sigma^z nd^{z/2}}{\mathsf{OPT}})$-coreset and an $(+\infty, \Omega(\frac{\sigma^z nd^{z/2}}{\mathsf{OPT}}))$-approximation-ratio coreset of of $P$ for $(k, z)$-CLUSTERING when $k = n - 1$.*

**Proof:** By a similar argument as that of Theorem E.1, we only need to prove the following property:

$$\mathbb{E}_{\widehat{P}}\left[\sum_{p \in P} \|\xi_p\|_2^2\right] = O_z(\sigma^z nd^{z/2}), \text{ and } \mathrm{Var}_{\widehat{P}}\left[\sum_{p \in P} \|\xi_p\|_2^2\right] = O_z(\sigma^{2z} nd^z),$$

which again holds by the Bernstein condition. $\qquad\square$

## F    MORE EMPIRICAL RESULTS

In this section, we show more empirical results to corroborate our findings in Section 4.

**More results on Adult dataset**   Figure 1 only shows the result on Bank dataset, and we show similar figure in Figure 2 on Adult dataset. In Table 1, we fix the noise level $\theta = 0.1$, and we also show similar table for $\theta = 0.08$ (Table 2) and $\theta = 0.8$ (Table 3). All our empirical findings in Section 4 still hold.

**Experiment with different $k$**   We also show experiment results for different number of clusters $k$ (Figure 4 and 3) on Bank dataset. All our empirical findings in Section 4 still hold.

**Data assumption results on real-world dataset**   We also test if the data assumption are satisfied on real-world datasets. For a dataset $\mathcal{D}$, we run the $k$-MEANS clustering algorithm on dataset $\mathcal{D}$ and get the centers $C$ and a partition of the dataset $\{P_1, \ldots, P_k\}$ where $P_i$ is clustered to $c_i \in C$. We compute $\mathsf{OPT}_i = \mathrm{cost}(P_i, c_i)$ and $a_i = \max_{p \in P_i} \|p - c_i\|$. We then compute

$$r_{\mathsf{OPT}} := \max_{i,j} \frac{\mathsf{OPT}_i}{\mathsf{OPT}_j}, \quad r_a := \max_{i,j} \frac{\|c_i - c_j\|}{a_i + a_j}.$$

Here $r_{\mathsf{OPT}}$ measures the maximum ratio between the cost of different clusters, and $r_a$ "upper bounds" how separated the dataset is.

Table 2: The empirical errors $\widehat{\text{Err}}(S)$ and $\widehat{\text{Err}}_{\text{AR}}(S)$ under different noise models, different coreset sizes, and different datasets (`Bank` and `Adult`). We fix the noise level $\theta = 0.08$. We repeat each setting 20 times, and we provide the mean and the standard deviation of the empirical errors.

(a) Results for `Bank` dataset.

| Coreset Size | Model I, Gaussian | | Model II, Gaussian | | Model I, Laplace | | Model II, Laplace | |
|---|---|---|---|---|---|---|---|---|
| | $\widehat{\text{Err}}(S)$ | $\widehat{\text{Err}}_{\text{AR}}(S)$ | $\widehat{\text{Err}}(S)$ | $\widehat{\text{Err}}_{\text{AR}}(S)$ | $\widehat{\text{Err}}(S)$ | $\widehat{\text{Err}}_{\text{AR}}(S)$ | $\widehat{\text{Err}}(S)$ | $\widehat{\text{Err}}_{\text{AR}}(S)$ |
| 500 | $0.195_{0.05}$ | $0.076_{0.02}$ | $0.199_{0.04}$ | $0.083_{0.03}$ | $0.216_{0.05}$ | $0.079_{0.02}$ | $0.192_{0.04}$ | $0.075_{0.02}$ |
| 1000 | $0.180_{0.02}$ | $0.045_{0.01}$ | $0.183_{0.03}$ | $0.044_{0.01}$ | $0.168_{0.03}$ | $0.050_{0.01}$ | $0.180_{0.02}$ | $0.047_{0.01}$ |
| 2000 | $0.166_{0.02}$ | $0.035_{0.01}$ | $0.166_{0.02}$ | $0.030_{0.01}$ | $0.159_{0.02}$ | $0.038_{0.01}$ | $0.170_{0.02}$ | $0.028_{0.01}$ |
| 3000 | $0.166_{0.02}$ | $0.027_{0.01}$ | $0.162_{0.01}$ | $0.024_{0.01}$ | $0.153_{0.02}$ | $0.025_{0.01}$ | $0.167_{0.02}$ | $0.022_{0.01}$ |
| 4000 | $0.151_{0.01}$ | $0.029_{0.01}$ | $0.162_{0.02}$ | $0.022_{0.01}$ | $0.151_{0.01}$ | $0.026_{0.01}$ | $0.159_{0.01}$ | $0.021_{0.01}$ |
| 5000 | $0.150_{0.01}$ | $0.026_{0.01}$ | $0.152_{0.01}$ | $0.017_{0.01}$ | $0.144_{0.01}$ | $0.025_{0.01}$ | $0.152_{0.01}$ | $0.024_{0.01}$ |

(b) Results for `Adult` dataset.

| Coreset Size | Model I, Gaussian | | Model II, Gaussian | | Model I, Laplace | | Model II, Laplace | |
|---|---|---|---|---|---|---|---|---|
| | $\widehat{\text{Err}}(S)$ | $\widehat{\text{Err}}_{\text{AR}}(S)$ | $\widehat{\text{Err}}(S)$ | $\widehat{\text{Err}}_{\text{AR}}(S)$ | $\widehat{\text{Err}}(S)$ | $\widehat{\text{Err}}_{\text{AR}}(S)$ | $\widehat{\text{Err}}(S)$ | $\widehat{\text{Err}}_{\text{AR}}(S)$ |
| 500 | $0.170_{0.04}$ | $0.053_{0.02}$ | $0.185_{0.04}$ | $0.063_{0.02}$ | $0.180_{0.04}$ | $0.057_{0.02}$ | $0.165_{0.04}$ | $0.063_{0.03}$ |
| 1000 | $0.166_{0.03}$ | $0.030_{0.02}$ | $0.156_{0.02}$ | $0.033_{0.02}$ | $0.160_{0.03}$ | $0.032_{0.02}$ | $0.159_{0.03}$ | $0.031_{0.01}$ |
| 2000 | $0.151_{0.02}$ | $0.012_{0.01}$ | $0.145_{0.02}$ | $0.017_{0.01}$ | $0.149_{0.03}$ | $0.014_{0.01}$ | $0.153_{0.02}$ | $0.018_{0.01}$ |
| 3000 | $0.143_{0.02}$ | $0.008_{0.01}$ | $0.148_{0.02}$ | $0.016_{0.01}$ | $0.145_{0.01}$ | $0.012_{0.01}$ | $0.145_{0.02}$ | $0.014_{0.01}$ |
| 4000 | $0.148_{0.02}$ | $0.005_{0.00}$ | $0.145_{0.01}$ | $0.007_{0.00}$ | $0.143_{0.02}$ | $0.005_{0.00}$ | $0.145_{0.01}$ | $0.010_{0.01}$ |
| 5000 | $0.145_{0.01}$ | $0.004_{0.00}$ | $0.144_{0.01}$ | $0.007_{0.01}$ | $0.142_{0.02}$ | $0.004_{0.00}$ | $0.148_{0.01}$ | $0.006_{0.00}$ |

Table 3: The empirical errors $\widehat{\text{Err}}(S)$ and $\widehat{\text{Err}}_{\text{AR}}(S)$ under different noise models, different coreset sizes, and different datasets (`Bank` and `Adult`). We fix the noise level $\theta = 0.8$. We repeat each setting 20 times, and we provide the mean and the standard deviation of the empirical errors.

(a) Results for `Bank` dataset.

| Size | Model I, Gaussian | | Model II, Gaussian | | Model I, Laplace | | Model II, Laplace | |
|---|---|---|---|---|---|---|---|---|
| | $\widehat{\text{Err}}(S)$ | $\widehat{\text{Err}}_{\text{AR}}(S)$ | $\widehat{\text{Err}}(S)$ | $\widehat{\text{Err}}_{\text{AR}}(S)$ | $\widehat{\text{Err}}(S)$ | $\widehat{\text{Err}}_{\text{AR}}(S)$ | $\widehat{\text{Err}}(S)$ | $\widehat{\text{Err}}_{\text{AR}}(S)$ |
| 500 | $1.589_{0.06}$ | $0.277_{0.04}$ | $1.605_{0.06}$ | $0.315_{0.05}$ | $1.586_{0.07}$ | $0.243_{0.04}$ | $1.611_{0.06}$ | $0.283_{0.04}$ |
| 1000 | $1.566_{0.06}$ | $0.169_{0.03}$ | $1.597_{0.06}$ | $0.201_{0.03}$ | $1.607_{0.06}$ | $0.153_{0.02}$ | $1.593_{0.05}$ | $0.178_{0.03}$ |
| 2000 | $1.580_{0.06}$ | $0.120_{0.02}$ | $1.580_{0.05}$ | $0.146_{0.02}$ | $1.597_{0.06}$ | $0.100_{0.01}$ | $1.578_{0.05}$ | $0.137_{0.02}$ |
| 3000 | $1.571_{0.05}$ | $0.102_{0.01}$ | $1.580_{0.06}$ | $0.128_{0.02}$ | $1.580_{0.05}$ | $0.090_{0.02}$ | $1.585_{0.04}$ | $0.102_{0.02}$ |
| 4000 | $1.583_{0.04}$ | $0.085_{0.01}$ | $1.582_{0.06}$ | $0.109_{0.01}$ | $1.576_{0.04}$ | $0.074_{0.01}$ | $1.587_{0.05}$ | $0.098_{0.02}$ |
| 5000 | $1.575_{0.04}$ | $0.081_{0.02}$ | $1.580_{0.05}$ | $0.102_{0.02}$ | $1.574_{0.04}$ | $0.069_{0.01}$ | $1.580_{0.05}$ | $0.089_{0.01}$ |

(b) Results for `Adult` dataset.

| Size | Model I, Gaussian | | Model II, Gaussian | | Model I, Laplace | | Model II, Laplace | |
|---|---|---|---|---|---|---|---|---|
| | $\widehat{\text{Err}}(S)$ | $\widehat{\text{Err}}_{\text{AR}}(S)$ | $\widehat{\text{Err}}(S)$ | $\widehat{\text{Err}}_{\text{AR}}(S)$ | $\widehat{\text{Err}}(S)$ | $\widehat{\text{Err}}_{\text{AR}}(S)$ | $\widehat{\text{Err}}(S)$ | $\widehat{\text{Err}}_{\text{AR}}(S)$ |
| 500 | $1.359_{0.09}$ | $0.335_{0.05}$ | $1.347_{0.08}$ | $0.419_{0.04}$ | $1.358_{0.12}$ | $0.274_{0.03}$ | $1.373_{0.07}$ | $0.344_{0.08}$ |
| 1000 | $1.365_{0.09}$ | $0.252_{0.04}$ | $1.309_{0.08}$ | $0.321_{0.05}$ | $1.349_{0.10}$ | $0.209_{0.04}$ | $1.352_{0.06}$ | $0.266_{0.04}$ |
| 2000 | $1.372_{0.08}$ | $0.210_{0.03}$ | $1.314_{0.05}$ | $0.286_{0.04}$ | $1.335_{0.07}$ | $0.162_{0.02}$ | $1.332_{0.07}$ | $0.195_{0.02}$ |
| 3000 | $1.357_{0.08}$ | $0.181_{0.02}$ | $1.331_{0.05}$ | $0.246_{0.02}$ | $1.345_{0.09}$ | $0.149_{0.01}$ | $1.345_{0.07}$ | $0.181_{0.01}$ |
| 4000 | $1.365_{0.09}$ | $0.170_{0.02}$ | $1.332_{0.05}$ | $0.236_{0.02}$ | $1.332_{0.08}$ | $0.145_{0.01}$ | $1.348_{0.05}$ | $0.172_{0.02}$ |
| 5000 | $1.376_{0.09}$ | $0.164_{0.02}$ | $1.325_{0.04}$ | $0.216_{0.03}$ | $1.340_{0.08}$ | $0.136_{0.01}$ | $1.355_{0.06}$ | $0.165_{0.01}$ |

We test $r_{\text{OPT}}$ and $r_a$ on `Adult` and `Bank` datasets, and test $k$-MEANS with $k = 5, 10, 15$. For each setting, we repeat the experiment with 10 times, and we report the mean and standard deviation in Table 4.

From the result, we can observe that the `Bank` dataset is indeed balanced, since $r_{\text{OPT}}$ is not very large, but the `Adult` dataset is not as balanced as `Bank` dataset. Besides, we find that the real-world `Adult` and `Bank` datasets are not well-separated, since $r_a$ is very small. This suggests that in reality, we might not need the data assumptions and the data assumption is only required for the current theoretical analysis.

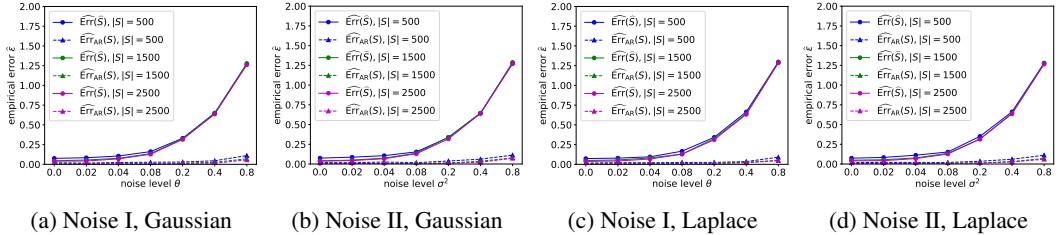

(a) Noise I, Gaussian     (b) Noise II, Gaussian     (c) Noise I, Laplace     (d) Noise II, Laplace

Figure 3: The empirical errors $\widehat{\mathrm{Err}}(S)$ and $\widehat{\mathrm{Err}}_{\mathrm{AR}}(S)$ versus the noise level $\theta$ plot for coreset $S$ with different sizes. The solid lines in each figure represent the coreset measure $\widehat{\mathrm{Err}}(S)$, and the dashed lines represent the weak coreset measure $\widehat{\mathrm{Err}}_{\mathrm{AR}}(S)$. We show the results from Bank dataset with 5 number of clusters, and Figures 3a to 3d denote the noise models I, II with Gaussian and Laplace noise respectively.

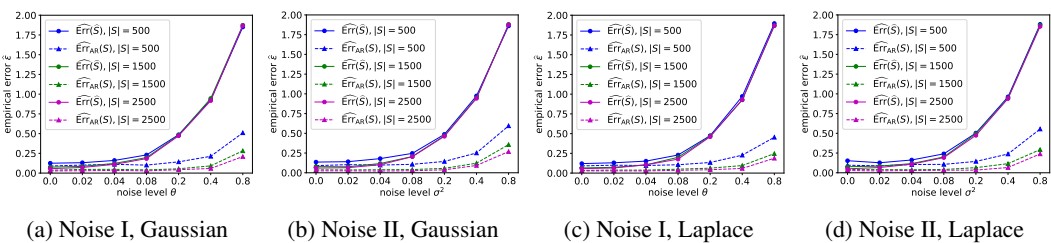

(a) Noise I, Gaussian     (b) Noise II, Gaussian     (c) Noise I, Laplace     (d) Noise II, Laplace

Figure 4: The empirical errors $\widehat{\mathrm{Err}}(S)$ and $\widehat{\mathrm{Err}}_{\mathrm{AR}}(S)$ versus the noise level $\theta$ plot for coreset $S$ with different sizes. The solid lines in each figure represent the coreset measure $\widehat{\mathrm{Err}}(S)$, and the dashed lines represent the weak coreset measure $\widehat{\mathrm{Err}}_{\mathrm{AR}}(S)$. We show the results from Bank dataset with 15 number of clusters, and Figures 4a to 4d denote the noise models I, II with Gaussian and Laplace noise respectively.

Table 4: $r_{\mathrm{OPT}}$ and $r_a$ on Adult and Bank datasets under different number of clusters $k$. For each $k$, we repeat the experiment for 10 times. We report the mean and standard deviation.

| Dataset | Adult | | | Bank | | |
|---|---|---|---|---|---|---|
| $k$ | $k=5$ | $k=10$ | $k=15$ | $k=5$ | $k=10$ | $k=15$ |
| $r_{\mathrm{OPT}}$ | $61.84_{6.44}$ | $16.26_{2.68}$ | $10.38_{2.22}$ | $6.71_{2.01}$ | $2.96_{0.44}$ | $3.75_{0.79}$ |
| $r_a$ | $1.26_{0.00}$ | $1.67_{0.01}$ | $1.94_{0.15}$ | $0.32_{0.05}$ | $0.59_{0.06}$ | $0.84_{0.04}$ |

**Total sensitivity** We also test how the total sensitivity Feldman & Langberg (2011) changes when the noise level ($\theta$ or $\sigma$) changes. We calculate the total sensitivity on Bank dataset under noise model I with Gaussian perturbation, with noise level ranging from $\gamma = 0$ to $\gamma = 0.2$. Table 5 summarizes the results. We find that the total sensitivity score decreases as the noise parameter $\theta$ increases. Intuitively, this phenomenon may be caused by adding larger noise drives points more "similar"

Table 5: total sensitivity on Bank dataset under noise model I with Gaussian perturbation, with noise level ranging from $\gamma = 0$ to $\gamma = 0.2$. We found that the larger the noise level, the smaller the total sensitivity.

| | total sensitivity($\times 10^5$) |
|---|---|
| $\theta = 0$ | 9.1 |
| $\theta = 0.02$ | 8.5 |
| $\theta = 0.04$ | 7.9 |
| $\theta = 0.08$ | 7.3 |
| $\theta = 0.2$ | 6.0 |

