# 1 ADDITIONAL EXPERIMENTS FOR CENSUS1990 DATASET

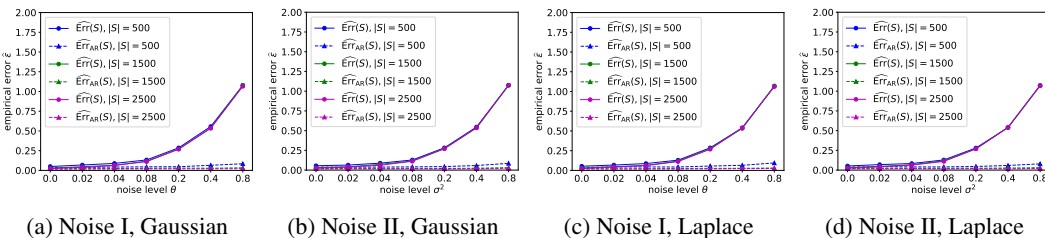

(a) Noise I, Gaussian    (b) Noise II, Gaussian    (c) Noise I, Laplace    (d) Noise II, Laplace

Figure 1: The empirical errors $\widehat{\mathrm{Err}}(S)$ and $\widehat{\mathrm{Err}}_{\mathrm{AR}}(S)$ versus the noise level ($\theta$ or $\sigma^2$) plot for coreset $S$ with different sizes. The solid lines in each figure represent the coreset measure $\widehat{\mathrm{Err}}(S)$, and the dashed lines represent the approximation-ratio measure $\widehat{\mathrm{Err}}_{\mathrm{AR}}(S)$. We show the results from Census1990 dataset, and Figures 1a to 1d denote the noise models I, II with Gaussian and Laplace noise respectively.

Table 1: Empirical errors $\widehat{\mathrm{Err}}(S)$ and $\widehat{\mathrm{Err}}_{\mathrm{AR}}(S)$ under different noise models and coreset sizes for Census1990 dataset. We fix the noise level $\theta = \sigma^2 = 0.2$. We repeat each setting 20 times, and provide the mean and the standard deviation (in the subscript) of the empirical errors.

| Size | Model I, Gaussian | | Model II, Gaussian | | Model I, Laplace | | Model II, Laplace | |
|------|-------------------|---|--------------------|---|------------------|---|-------------------|---|
| | $\widehat{\mathrm{Err}}(S)$ | $\widehat{\mathrm{Err}}_{\mathrm{AR}}(S)$ | $\widehat{\mathrm{Err}}(S)$ | $\widehat{\mathrm{Err}}_{\mathrm{AR}}(S)$ | $\widehat{\mathrm{Err}}(S)$ | $\widehat{\mathrm{Err}}_{\mathrm{AR}}(S)$ | $\widehat{\mathrm{Err}}(S)$ | $\widehat{\mathrm{Err}}_{\mathrm{AR}}(S)$ |
| 500 | $0.286_{0.02}$ | $0.047_{0.01}$ | $0.282_{0.01}$ | $0.044_{0.01}$ | $0.286_{0.02}$ | $0.055_{0.01}$ | $0.280_{0.02}$ | $0.046_{0.01}$ |
| 1000 | $0.275_{0.02}$ | $0.030_{0.01}$ | $0.275_{0.01}$ | $0.029_{0.01}$ | $0.275_{0.01}$ | $0.034_{0.01}$ | $0.276_{0.01}$ | $0.029_{0.01}$ |
| 2000 | $0.272_{0.01}$ | $0.022_{0.01}$ | $0.270_{0.01}$ | $0.019_{0.01}$ | $0.269_{0.01}$ | $0.021_{0.01}$ | $0.272_{0.01}$ | $0.022_{0.01}$ |
| 3000 | $0.262_{0.01}$ | $0.016_{0.01}$ | $0.271_{0.01}$ | $0.016_{0.00}$ | $0.271_{0.01}$ | $0.019_{0.01}$ | $0.268_{0.01}$ | $0.015_{0.01}$ |
| 4000 | $0.268_{0.01}$ | $0.017_{0.01}$ | $0.269_{0.01}$ | $0.015_{0.00}$ | $0.264_{0.01}$ | $0.017_{0.01}$ | $0.266_{0.00}$ | $0.012_{0.01}$ |
| 5000 | $0.263_{0.01}$ | $0.017_{0.01}$ | $0.265_{0.01}$ | $0.011_{0.01}$ | $0.266_{0.01}$ | $0.015_{0.01}$ | $0.267_{0.01}$ | $0.014_{0.01}$ |