# OpenReview forum: "Coresets for Clustering with Noisy Data"
_ICLR.cc/2024/Conference — Submitted to ICLR 2024_

### Official Review · Reviewer_a2UA · 2023-10-29

**Soundness:** 3 good
**Presentation:** 3 good
**Contribution:** 2 fair
**Rating:** 5
**Confidence:** 4

**Summary:**

The paper studies the problem of coreset for $k$-means clustering in the noisy setting.
Given a dataset $P \subset\mathbb{R}^d$ and a set $C$ of $k$ centers, one can define the cost to be $\sum_{x\in P} \min_{c\in C}\| x-c \|^2$.
The $k$-means clustering problem is to find the set $C$ that minimize the cost given a dataset $P$.
However, when the size of $P$ is huge, one common technique to improve the storage and computation is to extract a smaller subset $S$ of $P$ and find the set $C$ such that the $k$-means cost is minimized for $S$.
Different previous results showed that we can construct such a subset $S$ such that the size of $S$ is small and the difference between the $k$-means cost for $P$ and $S$ is also small.
In reality, the data is often noisy.
Therefore, we would like to construct a coreset from the noisy dataset $\widehat{P}$.
The authors showed that the standard notion of coreset for $k$-means is too strong for noisy dataset and defined a new notion for the noisy setting.
The authors then showed that if $S$ is a good coreset for $\widehat{P}$ under some mild assumptions on the noise then $S$ is also a good coreset for $P$.
The main idea is to first show that $\widehat{P}$ itself is a good coreset of $P$ even though the size is the same.
Then, using the composition property, the authors proved the main theorem.
Also, the authors provided some experimental results.

**Strengths:**

- The problem is well-motivated.
Most of the previous results did not consider the noisy version of the problem while the real world data is often noisy.
Hence, I believe it is a good starting point to investigate this line of work.

- The paper is well-written.
Readers of all levels of expertise should be able to understand this paper.

**Weaknesses:**

- Most techniques are straightforward calculations.
I am not sure if there are any fundamentally new techniques introduced in this paper.

**Questions:**

na

---

> ### Author Response · Authors · 2023-11-20
> **Response to Reviewer a2UA**
>
> Thanks for appreciating the motivation of our problem and the writing. As you noted, one of the main contributions of our paper is to initiate the study of coresets in the noisy setting.
>
> We think that establishing the separation between the two error measures $\mathrm{Err}$ and $\mathrm{Err}_{AR}$, both theoretically and empirically, is also interesting.
>
> Moreover, in formulating responses to other reviewers we have presented additional features/generalizations of our results that may be of interest.
>
> * In responses to Reviewers gA35 and QbwN, we provide an example to illustrate that, without assuming well-separated clusters, the two error measures $\mathrm{Err}$ and $\mathrm{Err}_{AR}$ may be the same; showing the necessity of the assumption of well-separated clusters.
> * Addressing the concern of Reviewer QbwN about the assumption of independent noise across dimensions, we illustrate how our techniques can be extended to handle non-independent noise across attributes of data points.
> * Based on the suggestion of Reviewer QbwN, we evaluate our results on the Census1990 dataset whose dimension is 68. We use the same setup as in Section 4 and the new results can be found in the following file in the supplementary material (https://openreview.net/attachment?id=D96juYQ2NW&name=supplementary_material). These empirical findings validate our theoretical results on datasets with a large number of attributes.

---

> > ### Comment · Reviewer_a2UA · 2023-11-22
> >
> > Thanks for the response. I will take it into consideration during the AC discussion.

---

### Official Review · Reviewer_QbwN · 2023-10-31

**Soundness:** 2 fair
**Presentation:** 2 fair
**Contribution:** 2 fair
**Rating:** 5
**Confidence:** 4

**Summary:**

This paper evaluates the performance of the optimal solution $\hat{C}$ in the original data $P$ when dealing with noisy data $\hat{P}$. The analysis is based on the coreset technique, but the traditional measure for determining coreset quality is too strong when the data is noisy. To address this issue, the authors propose the AR-coreset, which restricts itself to a local sublevel-region and considers the cost ratio. This new measure allows the authors to obtain refined estimates of $\hat{P}$. The authors demonstrate that $\hat{P}$ is a (1+nd)-coreset and a (1+kd)-AR-coreset of $P$, meaning that $\hat{C}$ is a (1+nd)-approximation and a (1+kd)-approximation solution, respectively.

**Strengths:**

1, The motivation is clear. It is instructive to consider the local sub-optimal region of the solution space in the AR-coreset.

2, Utilizing the coreset technique to analyze the approximation of the solution is an interesting approach.

3, The lower bound of the approximation is also discussed.

**Weaknesses:**

1. The failure of the 'Err()' seems natural since it considers the worst case in the entire solution space.
2. This paper does not provide a detailed comparison of the proposed method with other existing analysis methods for clustering with noisy data.
3. This paper considers independent noise across dimensions, the real noise might be correlated. The experiment part is too simple, which considers only two datasets; also the dimensions are low (6 and 10).
4. The noise model and the assumptions make the result to be relatively narrow.

**Questions:**

1, the authors say ‘Intuitively, how good a center set we can obtain from $\hat{P}$ is affected by the number of sign changes. ’. Please provide more thorough interpretations.

2, Besides determining the quality of a coreset in the presence of noise, are there other potential applications of the proposed AR-coreset?

3, Can we conclude that the (near) optimal solution(s) are robust to noise for other problems?

4, In the definition of AR-coreset, why should we consider the ratio of the cost for a given center set to the minimum cost? What would happen if we only consider the denominator of $r_P(\widehat{c})=\frac{\operatorname{cost}(P, \widehat{c})}{\mathrm{OPT}}$.

---

> ### Author Response · Authors · 2023-11-20
> **Response to Reviewer QbwN**
>
> Thanks for appreciating our motivations and results for the lower bound. We address your concerns and questions below.
>
> >1. The failure of the 'Err()' seems natural since it considers the worst case in the entire solution space.
>
> Thanks for raising this point. Even in the noise-free setting, $\mathrm{Err}(\cdot)$ considers the worst case in the entire solution space but, as the example below shows, it may not fail and capture the quality of the optimal center set of a coreset. The non-trivial contribution is in identifying a measure $\mathrm{Err}\_{AR}(\cdot)$ and presenting quantitative separation between two error measures $\mathrm{Err}(\cdot)$ and $\mathrm{Err}\_{AR}(\cdot)$ in the presence of noise.
>
> Consider the worst case instance constructed in [Cohen-Addad et al., 2022] where $P=\\{ e_1, e_2,\ldots, e_d \\} $ consists of unit basis vectors in $\mathbb{R}^{d}$ where $d=2k \varepsilon^{-2}$. Let $S\subset P$ be a (weighted) subset of size at most $k \varepsilon^{-2}$. The optimal center set of $S$ must lie on the spanned subspace of $S$, and hence can not be a $(1+\varepsilon)$-approximate center set of $P$. This geometric observation implies that $S$ is neither an $\varepsilon$-coreset nor a $(0,\varepsilon)$-approximation coreset. Then the tight sizes of both an $\varepsilon$-coreset and an $(0,\varepsilon)$-approximation coreset of such $P$ are $\Theta(k \varepsilon^{-2})$, which implies that $\mathrm{Err}(\cdot)$ and $\mathrm{Err}\_{AR}(\cdot)$ are the same on this dataset $P$.
>
> We will add this example in the final version.
>
> >2. This paper does not provide a detailed comparison of the proposed method with other existing analysis methods for clustering with noisy data.
>
> We have included a discussion of the literature on clustering with noisy data in Appendix A, along with a comparison of their analysis methods with our work. If you have any preferences regarding additional content or the placement of this section in the final version, please feel free to let us know -- we would be happy to do so.
>
> >3. This paper considers independent noise across dimensions, the real noise might be correlated.
>
> Thanks. We acknowledge that real noise may exhibit correlations across dimensions. As mentioned in the introduction, the attributes of the data may demonstrate weak correlations or interactions. For this type of real data, it is common to rely on the assumption of independent noise across dimensions, as observed in studies by [Zhu & Wu, 2004; Freitas, 2001; Langley et al., 1992].
>
> A positive aspect of our techniques is that they can be extended to handle non-independent noise across dimensions. Consider a scenario where the covariance matrix of each noise vector $\xi_p$ is $\Sigma\in \mathbb{R}^{d\times d}$ (with $\Sigma = \theta \cdot I_d$ when each $D_j = N(0,\theta)$ under the noise model II). Our proof of Theorem 3.1 relies on certain concentration properties of the terms $\sum\_{p\in P} \|\xi_P\|_2^2$ and $\sum\_{p\in P}\langle \xi_p, p-c\rangle$. Note that $\mathbb{E} \|\xi_p\|_2^2 = \mathrm{trace}(\Sigma)$. Hence, by a similar argument as in the proof of Claim 3.6, one can show that $\sum\_{p\in P} \|\xi_P\|_2^2$ concentrates on $n \cdot \mathrm{trace}(\Sigma)$ and $\sum\_{p\in P}\langle \xi_p, p-c\rangle \leq \|c-c^\star\|_2\cdot O(\sqrt{n\cdot \mathrm{trace}(\Sigma)})$. Consequently, an $\varepsilon$-coreset $S$ of $\widehat{P}$ is an $O(\varepsilon + \frac{n \cdot \mathrm{trace}(\Sigma)}{\mathrm{OPT}} + \sqrt{\frac{n \cdot \mathrm{trace}(\Sigma)}{\mathrm{OPT}}})$-coreset of $P$ and a $(0, O(\varepsilon + \frac{k \cdot \mathrm{trace}(\Sigma)}{\mathrm{OPT}}))$-approximation-ratio coreset of $P$. The only difference with Theorem 3.1 is that we replace the original variance term $\theta d$ to $\mathrm{trace}(\Sigma)$.
>
> We will add a remark on how our techniques can be extended to the non-independent noise case in the final version.

---

> > ### Comment · Reviewer_QbwN · 2023-11-23
> > **Thanks**
> >
> > Thanks for your response, and I will consider it in the next discussion stage.

---

> ### Author Response · Authors · 2023-11-20
> **Response continued**
>
> >4. The experiment part is too simple, which considers only two datasets; also the dimensions are low (6 and 10).
>
> Based on your comment, we evaluated our results on the Census1990 dataset whose dimension is 68. We use the same setup as in Section 4 and the new results can be found in the following file in the supplementary material (https://openreview.net/attachment?id=D96juYQ2NW&name=supplementary_material). These empirical findings validate our theoretical results on datasets with large number of attributes, including 1) a numerical separation between $\mathrm{Err}$ and $\mathrm{Err}\_{AR}$; 2) both $\mathrm{Err}$ and $\mathrm{Err}\_{AR}$ initially decrease and then stabilize as the coreset size increases; 3) variance of noise (or noise level) is the main parameter that determines both $\mathrm{Err}$ and $\mathrm{Err}\_{AR}$. Specifically, under the noise model II with Gaussian noise, $\mathrm{Err}(S)$ decreases from 0.282 to 0.270 as $|S|$ increases from 500 to 2000, and then remains around 0.270 as $|S|$ continues to increase to 5000. We will incorporate these results in the final version.
>
> >5. The noise model and the assumptions make the result to be relatively narrow.
>
> We have addressed your concern about the noise model in our response 3 above.
>
> Regarding the assumptions, in addition to the discussion in Section 3 explaining why both balancedness and well-separated clusters are reasonable and standard assumptions in the clustering literature, we provide an example below to illustrate that, without assuming well-separated clusters, the two error measures $\mathrm{Err}$ and $\mathrm{Err}\_{AR}$ may be the same.
>
> Consider a 3-Means problem over $\mathbb{R}$, i.e., $k=3$ and $d=1$. Let $P\subset \mathbb{R}$ consist of $\frac{n}{4}$ points each located at -1.01, -0.99, 0.99 and 1.01. A simple calculation shows that the optimal center set $C^\star$ is either $\\{-1.01, -0.99, 1\\}$ or $\\{-1, 0.99, 1.01\\}$, and $\mathrm{OPT} = 0.0005n$ (same in both cases).
>
> First note that, because there exist two clusters of points that are close to each other in both optimal center sets, $P$ does not satisfy the well-separateness assumption. (The distance between the nearby clusters is 0.02 while our assumption requires that it is at least $\sqrt{3}$.)
>
> Next, let $\widehat{P} \subset \mathbb{R}$ be an observed dataset drawn from $P$ under the noise model II with each $D_j = N(0,1)$. Since $\mathbb{E}\_{x\sim N(0,1)}[|x|\mid x<0] = \mathbb{E}\_{x\sim N(0,1)}[|x|\mid x\geq 0] = 1$, the optimal partition of $\widehat{P}$ is likely to be $\\{\widehat{p}: \widehat{p}\leq -1 \\}$, $\\{\widehat{p}: -1\leq \widehat{p}\leq 1 \\}$, $\\{\widehat{p}: \widehat{p}\geq 1\\}$ and the mean points of these three partitions are roughly -2, 0, 2 respectively. In this case, the optimal center set of $\widehat{P}$ is likely to be $\widehat{C} = \\{-2,0,2\\}$. Note that $\mathrm{cost}_2(P, \widehat{C}) \approx n \gg \mathrm{OPT}$. By Theorem E.1, we also have $\widehat{P}$ is a $\frac{n}{\mathrm{OPT}}$-coreset of $P$. Thus, $\mathrm{Err}(\widehat{P})\approx \mathrm{Err}\_{AR}(\widehat{P})\approx \frac{n}{\mathrm{OPT}}$.
>
> We will add this in the final version.

---

> ### Author Response · Authors · 2023-11-20
> **Response continued**
>
> >6. the authors say ‘Intuitively, how good a center set we can obtain from $\hat P$ is affected by the number of sign changes.’. Please provide more thorough interpretations.
>
> Thanks for your question. Below we explain how the quality of the optimal center set of $\hat P$ may be affected by the number of sign changes.
>
> Consider the case of 1-Means. Let $P \subset \mathbb{R}^d$ be an underlying dataset. Let $\widehat{P}_1$ and $\widehat{P}_2$ be observed datasets drawn from $P$ under the noise model II with noise parameters $\theta_1$ and $\theta_2$ respectively. We assume that $\theta_1 < \theta_2$. Recall that the optimal centers of $P$, $\widehat{P}_1$, and $\widehat{P}_2$ are denoted by $\mu(P)$, $\mu(\widehat{P}_1)$, and $\mu(\widehat{P}_2)$ respectively. We note that for any two centers $c$ and $c'$ lying on the line segment $\mu(P)-\mu(\widehat{P}_1)$, the signs of $\mathrm{cost}_2(P,c) - \mathrm{cost}_2(P,c')$ and $\mathrm{cost}_2(\widehat{P}_1,c) - \mathrm{cost}_2(\widehat{P}_1,c')$ must be different. This is besause the center that is closer to $\mu(P)$ has a smaller cost for $P$ but a larger cost for $\widehat{P}_1$ when compared to the other center. Hence, the number of sign changes from $P$ to $\widehat{P}_1$ is proportional to the distance $d(\mu(P), \mu(\widehat{P}_1))$. The same observation holds for $\widehat{P}_2$. Also, note that $d(\mu(P), \mu(\widehat{P}_1)) < d(\mu(P), \mu(\widehat{P}_2))$ since $\theta_1 < \theta_2$. Consequently, the number of sign changes from $P$ to $\widehat{P}_1$ is smaller than the number of sign changes from $P$ to $\widehat{P}_2$. Meanwhile, the quality of the optimal center $\mu(\widehat{P}_1)$ is $\mathrm{cost}_2(P, \mu(\widehat{P}_1)) = \mathrm{cost}_2(P, \mu(P)) + |P|\cdot d^2(\mu(P), \mu(\widehat{P}_1))$, which is better than the quality of $\mu(\widehat{P}_2)$ since $\mathrm{cost}_2(P, \mu(\widehat{P}_1)) < \mathrm{cost}_2(P, \mu(\widehat{P}_2))$.
>
> Hence, as the number of sign changes increases, the quality of the optimal center for the observed dataset deteriorates. We will add this example in the final version.
>
> >7. Besides determining the quality of a coreset in the presence of noise, are there other potential applications of the proposed AR-coreset?
>
> Thanks for asking this. Yes, the notion of AR-coreset may have potential applications in (noise-free cases of) machine learning tasks when one only wants to preserve near-optimal solutions, e.g., in regression. The notion of AR-coreset may be useful to further reduce the size of coreset. This is an interesting future direction and we will mention it in Section 5.
>
> >8. Can we conclude that the (near) optimal solution(s) are robust to noise for other problems?
>
> This is a great question. We have listed it as a future direction in Section 5. At present, the existing results seem insufficient to conclude the robustness of (near) optimal solutions to noise for other problems -- the answer is likely to be problem dependent.
>
> >9. In the definition of AR-coreset, why should we consider the ratio of the cost for a given center set to the minimum cost? What would happen if we only consider the denominator $r_P(\widehat{C})$.
>
> Since considering the denominator $r_P(\widehat{C})$ corresponds to a specific instance of the definition of AR-coreset (Definition 2.2) with $\alpha = 1$, using $r_P(\widehat{C})$ will only provide a guarantee that $S$ is a $(0, O(\varepsilon + \frac{\theta k d}{\mathrm{OPT}}))$-AR coreset of $P$. In contrast, considering the ratio $r_S(C)$ of the cost for a given center set $C$ corresponds to the case $\alpha \geq 1$, and enables a quantitative analysis of the affect of noise on center sets $C$ as we vary $\alpha$, for a given $S$.

---

### Official Review · Reviewer_gA35 · 2023-11-01

**Soundness:** 3 good
**Presentation:** 3 good
**Contribution:** 3 good
**Rating:** 6
**Confidence:** 4

**Summary:**

The paper studies the problem of computing coreset for a dataset from its noisy perturbation. The paper considers the most apparent approach: compute a coreset for the perturbed version, and use it directly as a coreset for the original dataset. The paper showed the following results:
1. The coreset from the noisy dataset can be very bad for the original dataset, in terms of the relative error Err commonly used to measure a coreset's approximation quality.
2. The authors notice that the traditional measure (the relative error mentioned in 1) is too strong because it is the supreme over all possible center sets, while in practice people is more interested on how well the coresets can approximate costs for "not so bad" center sets. This motivates the authors to design a new relative error $Err_{AR}$ (which they call "approximate error ratio") that only takes supreme only over center sets that approximates the optimal solution.
3. The authors show that this new definition of relative error can help give tigher approximation ratio estimation for center set computed on the coreset (obtained from the noisy dataset). In particular, a coreset $S$ with large Err can have much smaller $Err_{AR}$, which means a good approximate solution on $S$ will also have a small cost on the original dataset. While if we use $Err$ for estimation, the bound obtained is much looser.

**Strengths:**

I think the definition of $Err_{AR}$ is quite neat. The authors show that there is a strong separation between the traditional relative error $Err$ and their new measure $Err_{AR}$. I like this separation result in particular.

**Weaknesses:**

Although the authors claim the two assumptions (i.e. $O(1)$-balancedness and well-separation) are "mild" and only for "overcoming technical difficulty", I feel they are quite strong.
Also, one possibility is that the separation between $Err$ and $Err_{AR}$ is actually a result of these two assumptions. It would be great if the authors can show a quantitive analysis on the dependence between the separation and the two assumptions. For example, would it be possible that when the data become less balanced / well-separated, the two measures $Err$ and $Err_{AR}$ converge to each other. (My gut's feeling is the degree of well-separation could likely have a non-trivial effect on $Err$ / $Err_{AR}$). If that's the case, I feel this would somewhat strengthen the paper's result since $Err_{AR}$ can be viewed as a unifying measure that's tighter in extreme parameter ranges.

**Questions:**

There is no space between "$(k,z)$-Clustering" and the text following it. I guess the author should ad a `\xspace` after their macro for "$(k,z)$-Clustering"

---

> ### Author Response · Authors · 2023-11-20
> **Response to Reviewer gA35**
>
> Thanks for appreciating the novelty of our AR-notion and our results. We address your specific questions below.
>
> >The assumptions are quite strong ... one possibility is that the separation between $\mathrm{Err}$ and $\mathrm{Err}\_{AR}$ is actually a result of these two assumptions. It would be great if the authors can show a quantitive analysis on the dependence between the separation and the two assumptions...
>
> Thank you for suggesting to investigate this. Below we give an example that illustrates that if one does not assume that the clusters are well-separated, the two error measures $\mathrm{Err}$ and $\mathrm{Err}\_{AR}$ may be the same.
>
> Consider a 3-Means problem over $\mathbb{R}$, i.e., $k=3$ and $d=1$. Let $P\subset \mathbb{R}$ consist of $\frac{n}{4}$ points each located at -1.01, -0.99, 0.99, and 1.01. A simple calculation shows that the optimal center set $C^\star$ is either $\\{-1.01, -0.99, 1\\}$ or $\\{-1, 0.99, 1.01 \\}$, and $\mathrm{OPT} = 0.0005n$ (same in both cases).
>
> First note that, because there exist two clusters of points that are close to each other in both optimal center sets, $P$ does not satisfy the well-separateness assumption. (The distance between the nearby clusters is 0.02 while our assumption requires that it is at least $\sqrt{3}$.)
>
> Next, let $\widehat{P} \subset \mathbb{R}$ be an observed dataset drawn from $P$ under the noise model II with each $D_j = N(0,1)$. Since $\mathbb{E}\_{x\sim N(0,1)}[|x|\mid x<0] = \mathbb{E}\_{x\sim N(0,1)}[|x|\mid x\geq 0] = 1$, the optimal partition of $\widehat{P}$ is likely to be $\\{\widehat{p}: \widehat{p}\leq -1 \\}$, $\\{\widehat{p}: -1\leq \widehat{p}\leq 1 \\}$, $\\{\widehat{p}: \widehat{p}\geq 1 \\}$ and the mean points of these three partitions are roughly -2, 0, 2 respectively. In this case, the optimal center set of $\widehat{P}$ is likely to be $\widehat{C} = \\{-2,0,2\\}$. Note that $\mathrm{cost}\_2(P, \widehat{C}) \approx n \gg \mathrm{OPT}$. By Theorem E.1, we also have $\widehat{P}$ is a $\frac{n}{\mathrm{OPT}}$-coreset of $P$. Thus, $\mathrm{Err}(\widehat{P})\approx \mathrm{Err}\_{AR}(\widehat{P})\approx \frac{n}{\mathrm{OPT}}$.
>
> We will explain this formally in the final version.
>
>
> >There is no space between "$(k,z)$-Clustering" and the text following it. I guess the author should ad a \xspace after their macro for "$(k,z)$-Clustering"
>
> Thanks, we will fix it.

---

### Official Review · Reviewer_C3U6 · 2023-11-08

**Soundness:** 2 fair
**Presentation:** 2 fair
**Contribution:** 3 good
**Rating:** 6
**Confidence:** 3

**Summary:**

The paper shows how to construct coreset for clustering with noisy data.

**Strengths:**

- The paper introduces new measures that are used to construct coreset that guarantees eps approximations.

- The paper also presents a lower bound for 1-mean with noisy data.

- The theory is backed by a good number of empirical evaluations on real dataset.

**Weaknesses:**

- Some intuition and relation between the claims in section 2 will be helpful.

- A formal algorithm, even in the appendix, will increase its impact.

**Questions:**

- Please give an example of how the number of sign changes affects the goodness/quality of centers.

- Coreset size inversely proportional to n and d, in coreset measure is counter-intuitive. Comments?

---

> ### Author Response · Authors · 2023-11-20
> **Response to Reviewer C3U6**
>
> Thanks for appreciating our new measure and theoretical results. We address your questions below.
>
> >Comment: Some intuition and relation between the claims in Section 2 will be helpful.
>
> Thanks for the comment. The intuition behind Claim 2.3 is as follows: By definition, a $\beta$-approximate center set $C$ of $S$, for some $\beta \leq \alpha$, must be a $\beta(1+\varepsilon)$-approximate center set of $P$. This allows us to find a near-optimal center set of $P$ from $S$.
>
> This transitivity implies Claim 2.4 because we can obtain a near-optimal center set from a coreset or an approximation-ratio coreset $S$ of another coreset $S'$ of $P$. A bit more formally, Claim 2.3 ensures that an $\frac{\alpha}{1+O(\varepsilon)}$-approximation $C$ for $S$ is an $\alpha$-approximation for $S'$. Since $S'$ is an $\varepsilon'$-coreset of $P$, we conclude that $C$ is an $\alpha(1+\varepsilon')$-approximation for $P$.
>
> We will explain this in Section 2 in the final version.
>
> >Comment: A formal algorithm, even in the appendix, will increase its impact.
>
> It seems like there is some confusion. We do not propose a new coreset algorithm; instead, we show how any coreset algorithm can be used to construct a coreset in the presence of noise. In particular, Lemma 3.4 asserts that when $\varepsilon > \frac{\theta n d}{\mathrm{OPT}} + \sqrt{\frac{\theta n d}{\mathrm{OPT}}}$, one can apply a given coreset algorithm $A$ to construct an $\varepsilon$-coreset of $P$ given the noise parameter $\theta$.
>
> Specifically, when provided with the noise parameter $\theta$, the noisy dataset $\widehat{P}$, $k$, and $\varepsilon > \frac{\theta n d}{\mathrm{OPT}} + \sqrt{\frac{\theta n d}{\mathrm{OPT}}}$, one can simply return the coreset obtained from coreset algorithm $A$, where we set $P' = \widehat{P}$, $k' = k$, and $\varepsilon' = \varepsilon - \frac{\theta n d}{\mathrm{OPT}} + \sqrt{\frac{\theta n d}{\mathrm{OPT}}}$.
>
> The lemma also implies that when $\varepsilon \leq \frac{\theta n d}{\mathrm{OPT}} + \sqrt{\frac{\theta n d}{\mathrm{OPT}}}$, it is not possible to construct an $\varepsilon$-coreset of $P$.
>
> We will clarify this in the final version.
>
> >Please give an example of how the number of sign changes affects the goodness/quality of centers.
>
> Thanks for bringing this up. Below we explain how the quality of the optimal center set of $\hat P$ may be affected by the number of sign changes.
>
> Consider the case of 1-Means. Let $P \subset \mathbb{R}^d$ be an underlying dataset. Let $\widehat{P}_1$ and $\widehat{P}_2$ be observed datasets drawn from $P$ under the noise model II with noise parameters $\theta_1$ and $\theta_2$ respectively. We assume that $\theta_1 < \theta_2$. Recall that the optimal centers of $P$, $\widehat{P}_1$, and $\widehat{P}_2$ are denoted by $\mu(P)$, $\mu(\widehat{P}_1)$, and $\mu(\widehat{P}_2)$ respectively. We note that for any two centers $c$ and $c'$ lying on the line segment $\mu(P)-\mu(\widehat{P}_1)$, the signs of $\mathrm{cost}_2(P,c) - \mathrm{cost}_2(P,c')$ and $\mathrm{cost}_2(\widehat{P}_1,c) - \mathrm{cost}_2(\widehat{P}_1,c')$ must be different. This is besause the center that is closer to $\mu(P)$ has a smaller cost for $P$ but a larger cost for $\widehat{P}_1$ when compared to the other center. Hence, the number of sign changes from $P$ to $\widehat{P}_1$ is proportional to the distance $d(\mu(P), \mu(\widehat{P}_1))$. The same observation holds for $\widehat{P}_2$. Also, note that $d(\mu(P), \mu(\widehat{P}_1)) < d(\mu(P), \mu(\widehat{P}_2))$ since $\theta_1 < \theta_2$. Consequently, the number of sign changes from $P$ to $\widehat{P}_1$ is smaller than the number of sign changes from $P$ to $\widehat{P}_2$. Meanwhile, the quality of the optimal center $\mu(\widehat{P}_1)$ is $\mathrm{cost}_2(P, \mu(\widehat{P}_1)) = \mathrm{cost}_2(P, \mu(P)) + |P|\cdot d^2(\mu(P), \mu(\widehat{P}_1))$, which is better than the quality of $\mu(\widehat{P}_2)$ since $\mathrm{cost}_2(P, \mu(\widehat{P}_1)) < \mathrm{cost}_2(P, \mu(\widehat{P}_2))$.
>
> Hence, as the number of sign changes increases, the quality of the optimal center for the observed dataset deteriorates. We will add this example in the final version.

---

> ### Author Response · Authors · 2023-11-20
> **Response continued**
>
> >Coreset size inversely proportional to $n$ and $d$, in coreset measure is counter-intuitive. Comments?
>
> Thanks for your question. Below we explain why this is not counter-intuitive.
>
> First, the coreset size suggested by Lemma 3.4 contains an $\mathrm{OPT}^2$ term in the factor $\frac{\mathrm{OPT}^2}{n^2 d^2}$. $\mathrm{OPT}$ typically increases linearly with $n$, e.g., in the example in Appendix B,  $\mathrm{OPT} = n$. Thus,  $\frac{\mathrm{OPT}^2}{n^2 d^2}$ typically scales with $\frac{1}{d^2}$.
>
> Second, by Theorem 3.3, the error measure $\mathrm{Err}(\widehat{P}) \geq \Omega(\frac{\theta nd}{\mathrm{OPT}})$, which increases as $d$ increases. Thus, Lemma 3.4 implies that we cannot achieve an $\varepsilon$-coreset of $P$ for $\varepsilon \leq  \frac{\theta nd}{\mathrm{OPT}}$. This is different from the noise-free setting where we can achieve an $\varepsilon$-coreset for any $\varepsilon > 0$. Since a coreset algorithm usually constructs a coreset of size proportional to $\varepsilon^{-2}$, an $\varepsilon = \frac{\theta nd}{\mathrm{OPT}}$-coreset of $P$ is typically of size inversely proportional to $d^2$.
>
> For instance, applying the algorithm of [Cohen-Addad et al., 2022], one can construct an $\frac{\theta nd}{\mathrm{OPT}}$-coreset of $P$ with size $\tilde{O}(\frac{k^{1.5}}{\theta^2 d^2})$ when $\mathrm{OPT} \approx n$. We will explain this in the final version.

---

### Meta-Review · Area_Chair_wV86 · 2023-12-06

**Metareview:**

The paper develops a new notion of error that focuses only on the approximation factor of nearly optimal solutions instead of the error in strong coresets where one cares about the approximation factor for all solutions. It shows that in the case of noisy datasets, the traditional error for strong coresets is too strict, leading to huge and pessimistic approximation factors while the new notion remains meaningful. The paper shows that for datasets with clusters that are well-separated and balanced, any coreset for the noisy dataset is still a "good" coreset for the original dataset where the goodness depends on the amount of noise.

On the other hand, the main result on the strength of the new notion, tightening from strong coresets, relies on two strong assumptions, well-separation and balance. These assumptions are strong and might not hold for real datasets. Additionally, one reviewer suggests that there is no new coreset algorithm for the noisy setting, making it less clear whether some new algorithms can do better than applying any existing coreset construction. One reviewer suggests that the noise model is somewhat stylized and restrictive (independent noise across coordinates with bounded moments).

**Justification For Why Not Higher Score:**

The results are good to know but seem be somewhat restrictive. They only apply to well-separated and balanced clusters. Furthermore, the results are generic for all coreset constructions. They do not seem to rule out specific algorithms or other weaker notions such as weak coresets.

**Justification For Why Not Lower Score:**

N/A

---

### Decision · Program_Chairs · 2024-01-16

Reject